# Ecology-guided prediction of cross-feeding interactions in the human gut microbiome

Akshit Goyal[1,4], Tong Wang[2,4], Veronika Dubinkina[3] & Sergei Maslov [2,3✉]

Understanding a complex microbial ecosystem such as the human gut microbiome requires information about both microbial species and the metabolites they produce and secrete. These metabolites are exchanged via a large network of cross-feeding interactions, and are crucial for predicting the functional state of the microbiome. However, till date, we only have information for a part of this network, limited by experimental throughput. Here, we propose an ecology-based computational method, GutCP, using which we predict hundreds of new experimentally untested cross-feeding interactions in the human gut microbiome. GutCP utilizes a mechanistic model of the gut microbiome with the explicit exchange of metabolites and their effects on the growth of microbial species. To build GutCP, we combine metagenomic and metabolomic measurements from the gut microbiome with optimization techniques from machine learning. Close to 65% of the cross-feeding interactions predicted by GutCP are supported by evidence from genome annotations, which we provide for experimental testing. Our method has the potential to greatly improve existing models of the human gut microbiome, as well as our ability to predict the metabolic profile of the gut.

[1] Physics of Living Systems, Department of Physics, Massachusetts Institute of Technology, Cambridge, MA, USA. [2] Department of Physics, University of Illinois at Urbana-Champaign, Urbana, IL, USA. [3] Department of Bioengineering and Carl R. Woese Institute for Genomic Biology, University of Illinois at Urbana-Champaign, Urbana, IL, USA. [4] These authors contributed equally: Akshit Goyal, Tong Wang. ✉email: maslov@illinois.edu

The gut microbiome plays an important role in human health, and the ability to manipulate it holds immense potential to prevent and treat multiple diseases[1–8]. The microbiome comprises not only hundreds of microbial species but also hundreds of metabolites that they consume and secrete: a phenomenon called cross-feeding[9,10]. These metabolites—through which gut microbes interact with each other—mediate interspecies interactions and can even directly impact the host[11–14].

Indeed, metabolite levels in the gut are often more predictive of host health than species levels[11,15,16]. Therefore, developing a complete understanding of both the human gut microbiome together with the metabolome is necessary to positively control and manipulate human health.

A promising framework to realize such an understanding is a fully mechanistic model of the microbiome[17–20], which can connect the levels of microbial species and metabolites with each other quantitatively. An essential first step in building this model is establishing which metabolic interactions are relevant in the human gut microbiome[18,20–22]. Indeed, inferring cross-feeding interactions is an active and important field of microbiome research, and employs both direct[9,23–25] and indirect[11,26–29] inference methods. Direct methods, which comprise experimental verification of the metabolic activity of gut microbes, are slow, require painstaking effort, and thus miss many relevant interactions (i.e., they are incomplete). Indirect methods, which chiefly comprise inferring the metabolic activity of gut microbes from their genome sequences, are noisy, lack curation, and vastly overestimate relevant cross-feeding interactions (i.e., they are "beyond complete"; since they are based on genome annotations, they comprise both active and inactive interactions in the gut)[29–33]. We thus need new methods that represent the middle ground between direct and indirect methods. Specifically, we need methods that can use the directly inferred but incomplete interactions as a "bootstrap", allowing one to filter out the indirectly inferred but noisy ones. We believe that ecological consumer-resource models provide the means to perform this bootstrapping and predict new and ecologically sound cross-feeding interactions. Moreover, we believe that these methods can benefit from advances in machine learning[34,35], which is effective at identifying patterns in known data and using them to make new predictions.

Here, we propose GutCP, short for gut cross-feeding predictor: a new, general, and ecology-guided method to infer and predict cross-feeding interactions in the human gut microbiome. GutCP combines machine-learning techniques[34,35] with an ecological model of the microbiome. The ecological model is effective at bootstrapping previously known direct interactions and estimating the metabolic environment of the gut in agreement with experimental measurements[20]. GutCP uses these estimates as leverage to predict new cross-feeding interactions. The machine-learning techniques that GutCP employs help optimize and curate the process of inferring new interactions. We find that close to 65% of the interactions we predict are supported by the available genomic evidence. Our predictions can be easily tested by simple experiments, and have the potential to enable a fully mechanistic understanding of the human gut microbiome going beyond the analysis of correlations between species and metabolites.

## Results

### Overview of the GutCP algorithm.
Our approach uses the idea that we can leverage cross-feeding interactions—which comprise knowing the metabolites that each microbial species is capable of consuming and producing—to mechanistically connect the levels of microbes and metabolites in the human gut. Several different mechanistic models in past studies have shown that this is indeed possible[18,20,29,36,37]. While GutCP is generalizable and can be used with any of these models, in this paper, we use a previously published consumer-resource model[20]. We use this model because of its context and performance: it is built specifically for the human gut and is best able to explain the experimentally measured species composition of the gut microbiome with its resulting metabolic environment, or fecal metabolome (compared with other state-of-the-art methods, such as ref. [29]). To predict the metabolome from the microbiome, it relies on a manually curated set of known cross-feeding interactions[9]. It then uses these known interactions to follow the stepwise flow of metabolites through the gut. At each step (ecologically, at each trophic level), the metabolites available to the gut are utilized by microbial species that are capable of consuming them, and a fraction of these metabolites are secreted as metabolic byproducts. These byproducts are then available for consumption by another set of species in the next trophic level. After several such steps, the metabolites that are left unconsumed constitute the fecal metabolome.

We hypothesized that adding new, yet-undiscovered cross-feeding interactions would improve our ability to predict the levels of metabolites with our mechanistic and causal model. Specifically, we predict that the set of undiscovered interactions resulting in the most accurate and optimal improvement in predictions would be the most likely candidates for true cross-feeding interactions. Inferring such an optimal set of new cross-feeding interactions or reactions is the main logic driving GutCP. In what follows, we sometimes refer to cross-feeding reactions (i.e., metabolite consumption or production by microbes) as "links" in an overall cross-feeding network of the gut microbiome, whose nodes are microbes and metabolites (Fig. 1a; metabolites in blue, microbes in orange); the links themselves are directed edges connecting the nodes. Links can be of two types: consumption or nutrient uptake reactions (from nutrients to microbes) and production or nutrient secretion reactions (from microbes to their metabolic byproducts).

The salient aspects of our method are outlined in Fig. 1. We start with the known set of consumption and production links that were originally used by the model; these links are known from direct experiments and represent a ground-truth dataset or original cross-feeding network[9]. These are shown in Fig. 1a through the pink and blue arrows connecting nutrients 1 through 6 with microbes (a) through (c). For each sample, using only the species abundance from the microbiome, we use the model to quantitatively estimate the microbiome's species and metabolomic composition. Briefly, we assume that a defined set of polysaccharides, common to human diets, are available as the nutrient intake to the gut (nutrients 1 and 4 in Fig. 1a). We calculate the microbiome and metabolome profiles separately for each individual, which contain a different set of microbial species in their guts. At the first trophic level, all microbial species that are capable of using the polysaccharides (indicated by the pink arrows in Fig. 1a) consume each of them in proportion to their abundances (microbes a, b, and c in Fig. 1a). They subsequently secrete a fixed fraction of the consumed nutrients as metabolic byproducts; every species at this trophic level secretes all the metabolic byproducts it is known to secrete (blue arrows in Fig. 1a) in equal proportion (nutrients 2–6 in Fig. 1a). At the next trophic level, all species detected in the individual's gut which can consume the newly secreted byproducts consume them as nutrients, secreting a new set of byproducts, and this continues for four trophic levels (not shown in Fig. 1a for simplicity). At the end of this process, all metabolites which remain unconsumed by the community comprise the metabolome of the individual and the microbial species which consume nutrients and grow comprise the microbiome of the individual (for a complete description, see "Methods" and previous work[20]).

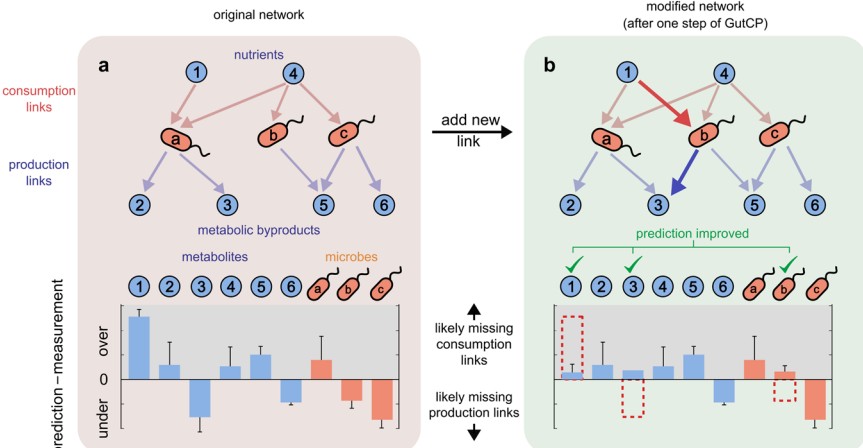

**Fig. 1 Overview of the GutCP algorithm. a** Schematic of the original set of known cross-feeding interactions (top) and bar plot of the prediction error for each metabolite and microbe (bottom). The cross-feeding interactions are represented as a network, whose nodes are either metabolites (cyan circles) or microbial species (orange ellipses), and directed links represent the abilities of different species to consume (red arrows) and produce (blue arrows) individual metabolites. **b** GutCP adds a new consumption link (red) and production link (blue) as added links reduce the prediction errors for metabolites and microbes.

For each metabolite and microbial species, there can be two kinds of prediction errors, or biases: individual (the sample-specific difference between predicted and measured levels) and systematic (average difference across all samples). We focused on the "systematic bias" for each metabolite and microbial species: the average deviation of the predicted levels from the measured levels across all samples in our dataset (Fig. 1a, bottom). The systematic bias for each metabolite and microbe tells us whether our model generally tends to predict their level to be greater than observed (overpredicted), less than observed (underpredicted), or neither (well-predicted). We assume that metabolites and microbes with a large systematic bias are most likely to harbor missing consumption or production links that are relevant across many samples. We prioritize adding links to them in proportion to their systematic biases.

After measuring the systematic bias for each metabolite and microbe, GutCP proceeds in discrete steps (Fig. 1a, b). At each step, we attempt to add a new link to the current cross-feeding network. This new link is chosen randomly from the entire set of combinatorially possible links (see "Methods"; for $S$ species, $M$ metabolites, and two kinds of links (consumption and production), there are a total of $2SM$ combinatorially possible links). We accept this link—keeping it in the current network—if it leads to an overall improvement in the agreement between the predicted and measured levels of microbes and metabolites. We repeat the process of adding new links—accepting or rejecting them—until the improvements in the levels of metabolites and microbes became insignificant. Overall, GutCP can add several links to improve the agreement between the predicted and measured levels of microbes and metabolites (in Fig. 1a, b, bottom, adding the extra red and blue link at the top results in improved predictions for metabolite (1), metabolite (3), and microbe (b). Figure 2a shows how the cross-feeding network improves over a typical GutCP run via the red trajectory, starting from the original network (Fig. 2a, top left) to the final network state (Fig. 2a, bottom right). Trajectories from 100 other runs are shown in gray. GutCP repeatedly reduces both the error of the metabolome predictions (y axis; measured as $\log_{10}\left(\frac{\text{pred}-\text{meas}}{\text{measurement}}\right)$) and improves the correlation between the predicted and measured metabolomes (x axis).

**Cross-validating the newly predicted interactions**. To test if the cross-feeding interactions predicted by GutCP are generalizable to unknown datasets, we performed fourfold cross-validation. We used a sample -omics dataset of the gut microbiome and metabolome sampled from 41 human individuals, comprising 221 metabolites and 72 microbial species (data from ref. [38]). We split our -omics dataset into two subsets: training (three-fourths of the individuals) and test (one-fourth of the individuals) subsets. We then ran GutCP on the training subset to discover new interactions and added them to the ground-truth interactions taken from ref. [9]. Doing so resulted in a network of cross-feeding interactions learned only from the training subset of the data. Finally, we evaluated the improvement in accuracy of metabolome predictions resulting from the trained network on the unseen, test subset of the data. We repeated this process three times, each time splitting the full dataset into a training subset (with a randomly chosen three-fourths of the individuals) and test subset (with the remaining one-fourth of the individuals); finally, we calculated the average improvement in prediction accuracy over all four splits.

We found that both the training and test set performances after using the links predicted by GutCP were significantly better than the baseline given by the original cross-feeding network (Table 1). Specifically, both measures of model performance, namely the logarithmic error and the average correlation, improved by 64% and 20%, respectively, after adding GutCP's discovered interactions. In addition, the test set performance was comparable to the training set performance (6% difference; Table 1). This suggests that the cross-feeding interactions inferred by GutCP are not likely to be a result of over-fitting.

**Building a consensus-based atlas of predicted cross-feeding interactions**. Having confirmed that GutCP is unlikely to over-fit data, we pooled the entire sample dataset of 41 individuals and ran 100 independent instances of our prediction algorithm on it; we verified that incorporating more instances did not qualitatively affect our results (Fig. 2b shows a rarefaction curve, which highlights the number of new links discovered by GutCP as we perform more runs the algorithm). Each run of the algorithm resulted in an average of 140 newly predicted cross-feeding interactions. Then, based on consensus from many runs, we assigned a confidence level to each predicted interaction, namely what fraction of GutCP runs it was discovered in. By calculating a null distribution (Fig. 2c, black), which predicts the fraction of

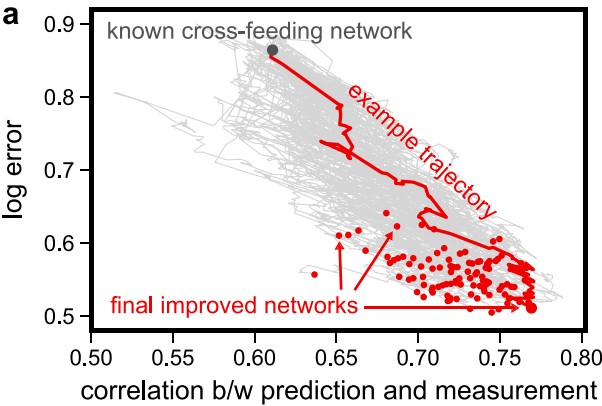

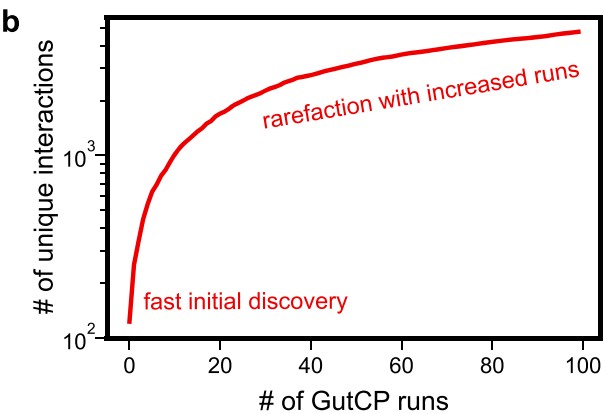

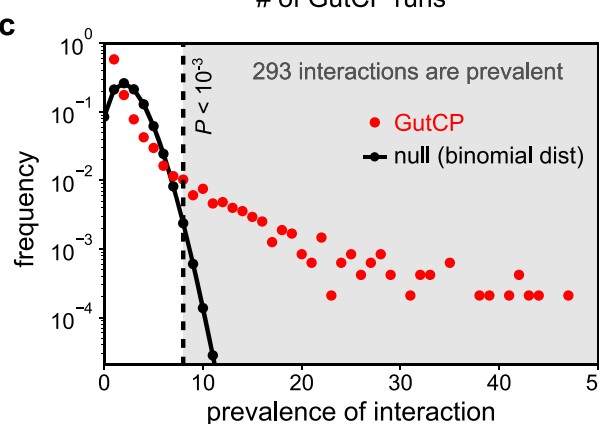

**Fig. 2 Improvement in predictions using GutCP. a** Improvement in log error ($\log_{10}(\frac{pred-meas}{measurement})$) and the correlation between the prediction and measured fecal metabolome during 100 typical runs of the GutCP algorithm. The gray point at the top left indicates the performance of the original cross-feeding network of Ref. [9], and the black points at the bottom right, that of improved networks predicted using GutCP. A trajectory example, highlighting how performance improves over a GutCP run, is shown in red, and others are shown in gray. **b** Rarefaction curve showing the number of unique cross-feeding interactions discovered by GutCP over 100 runs of the algorithm. **c** Prevalence of links, i.e., the number of GutCP runs in which they repeatedly appeared (red dots; total 100 runs) and for comparison, a corresponding binomial distribution with the same mean (black dotted line). P values for different prevalences are estimated using the one-sided binomial test.

GutCP runs where a random link would be discovered by chance, we assigned a P value to each link and set a threshold at $P = 10^{-3}$ (Fig. 3c, red; see "Methods" for details). Doing so finally resulted in a complete consensus-based atlas of 293 predicted cross-

**Table 1 Cross-validating the newly predicted interactions.**

|  | Metabolome pred — exp | Log error | Number of predicted metabolites |
|---|---|---|---|
| Original set | 0.61 | 0.89 | 17 |
| Training set | 0.72 ± 0.03 | 0.54 ± 0.02 | 30 ± 3 |
| Test set | 0.68 ± 0.04 | 0.59 ± 0.04 | 30 ± 3 |

Table showing the performance of the ecological cross-feeding model with the original set of interactions (consumption and production links), and with the additional interactions predicted by GutCP (both the training and test set performances; see main text). Performance is measured using three metrics: (1) the correlation between the predicted and experimentally measured metabolome, (2) the log error (see main text), and (3) the number of metabolites in the measured metabolome predicted by our ecological consumer-resource model. Values indicate mean for (1) and (2), and median for (3); errors, where shown, indicate standard deviation.

feeding interactions, which we have provided as a resource for experimental verification in Supplementary Table 1. Figure 3a shows a condensed version of these interactions obtained from the simulation with the best performance (the trajectory example in Fig. 2a with the lowest log error and highest correlation coefficient) in the form of a matrix; specifically, newly added interactions are in dark colors, and old interactions in faded colors. Supplementary Fig. 3 shows a complete version of this matrix. Note that some of the predicted interactions in Fig. 3a are unrealistic, e.g., the production of certain sugars like D-Fructose and D-Sorbitol. Such interactions are unlikely to be predicted in repeated simulations, and thus will not be part of the final consensus set. This illustrates the power of pooling results from several simulations to arrive at a set of highly probable predictions.

Network visualization of the complete consensus-based atlas of 293 predicted cross-feeding interactions is shown in Fig. 3b. Figure 3b also shows that the network of new interactions has two clear types of bacteria: on the left are "producers" and on the right are "consumers". We classified producers and consumers based on the directionality of the predicted interactions. *Bacteroides*, *Ruminococcus*, and *Bifidobacteria* are known byproduct producers in the gut microbiome[14,39–42], and as expected, GutCP predicted more production links for species in these genera. Known byproduct consumers, on the other hand (right of Fig. 3b), typically occupy the lower trophic levels, and our model originally underpredicted their abundances. Reasonably, GutCP added several new consumption links to them, allowing these species increased growth and accurately predicted abundances. Finally, some metabolites, like amino acids (e.g., L-alanine, L-tyrosine, and L-asparagine), short-chain fatty acids (e.g., propanoate, valerate, and butyrate) were predicted by GutCP to be mostly produced, not consumed, consistent with the literature[41,43].

**Large-scale effects and patterns observed in the human gut microbiome**. Equipped with our set of predicted cross-feeding interactions (production and consumption links), we examined the extent to which they affected and improved our model's predictions of the microbe and metabolite levels in the human gut microbiome. We found this improvement indeed significant. For a representative example, see Fig. 4a–d. Here, each panel compares the levels of microbes (Fig. 4a, b) or metabolites (Fig. 4c, d) predicted by the model (x axis) with the experimentally measured levels (y axis); the closer a point is to the marked line (indicating an exactly correct prediction), the better our predictive power. Even by visual inspection, one can see that the newly predicted links bring the points much closer to the line of correct predictions.

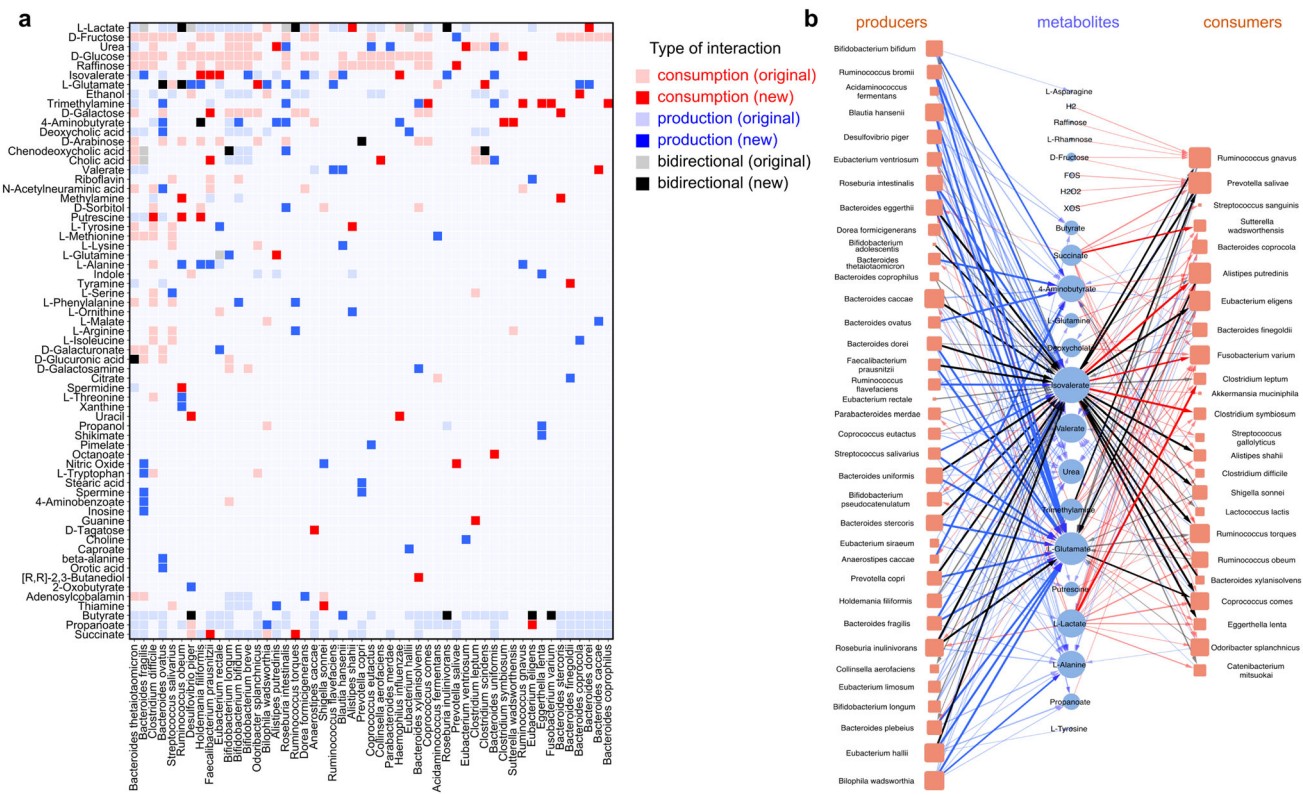

**Fig. 3 New cross-feeding interactions predicted by GutCP. a** Concise matrix representation of the improved cross-feeding network of the gut microbiome predicted by GutCP (the trajectory example in Fig. 2a with the best performance). The rows are metabolites, and columns, microbial species. Faded cells represent the original, known set of cross-feeding interactions, both production (light blue), consumption (light red), and bidirectional links (gray). The new cross-feeding interactions predicted by GutCP are shown in dark colors: production links in dark blue, consumption links in dark red, and bidirectional links in black. **b** Network of 293 new links predicted by GutCP (with a $P$ value $< 10^{-3}$, one-sided binomial test) during 100 independent simulations. Blue nodes represent metabolites, orange is bacteria as in Fig. 1. The size of each node represents its degree. The color of the links is the same as in (**a**), while the color intensity and link thickness are proportional to the link's confidence, or $P$ value. For bidirectional links, we represent the direction as that of the link with the smaller $P$ value.

By adding new cross-feeding links, GutCP nearly doubles the number of metabolites whose levels we could predict (roughly 30 metabolites, in contrast with 17 according to the original cross-feeding network; see Table 1). Namely, GutCP allows microbes to produce new metabolites that could not be produced according to the original cross-feeding network. As expected, these newly produced metabolites were indeed part of the experimentally measured metabolomes for these samples. Encouragingly, GutCP could predict their levels with an accuracy comparable with the original set of metabolites (compare Fig. 4d with Fig. 4c). Similarly, GutCP increased the number of microbial species whose levels we could predict. This was especially true of those microbial species, which could not grow given the original interactions (left-most points in Fig. 4a). By inferring the appropriate consumption links for these species, GutCP could also predict their levels correctly (in Fig. 4b, the left-most points moved close to the line of exact predictions).

Because our model mechanistically connects the abundances of microbes and metabolites, we next sought to understand how GutCP enabled such an improvement in the model's performance. We did this by comparing the change in the prediction error (or systematic bias) of each metabolite (Fig. 4e, white background; blue boxes indicate the original predictive error, and red boxes indicate the predictive error after adding GutCP's predictions) with the links that were added for each metabolite (Fig. 3).

We found that the newly predicted interactions had both direct and indirect effects on metabolite levels, and these were crucial for accurate prediction. By direct effects, we mean the following: if a systematically overpredicted (or underpredicted) metabolite was fixed by GutCP by inferring that a new microbe could consume (or produce) it, this new consumption (or production) link had a direct role in that metabolite's accurate prediction. For instance, we noticed that originally, spermidine was overpredicted (Fig. 4e, spermidine in blue); GutCP inferred a new consumption interaction (by *Ruminococcus obeum*; Fig. 3), and this corrected the spermidine level in the metabolome (Fig. 4e, spermidine in red), leading to a direct accurate prediction. Similarly, the amino acid lysine was underpredicted, which was fixed due to GutCP inferring a new production link (by *Blautia hansenii*; Fig. 3). Sometimes, a metabolite's under- or over-prediction was fixed as a result of GutCP inferring multiple consumptions or production links by several different microbial species in tandem (such as for putrescine, for which GutCP inferred three consumption links; Fig. 3). With only a subset of the inferred links, the levels of such metabolites still remained under- or overpredicted (on average, by one order of magnitude). Strikingly, we also observed several indirect effects of GutCP's predictions. Indirect effects comprise any discovered links where GutCP improves the prediction for a metabolite without adding a link that produces or consumes it. The improvement in prediction comes entirely from other added links, which can increase or decrease the levels of microbes that produce (or consume) that metabolite. For example, GutCP inferred no new consumption or production links for 5-aminovalerate (no predicted interactions in Fig. 3), but adding other links (e.g., the consumption of putrescine by *Clostridium*

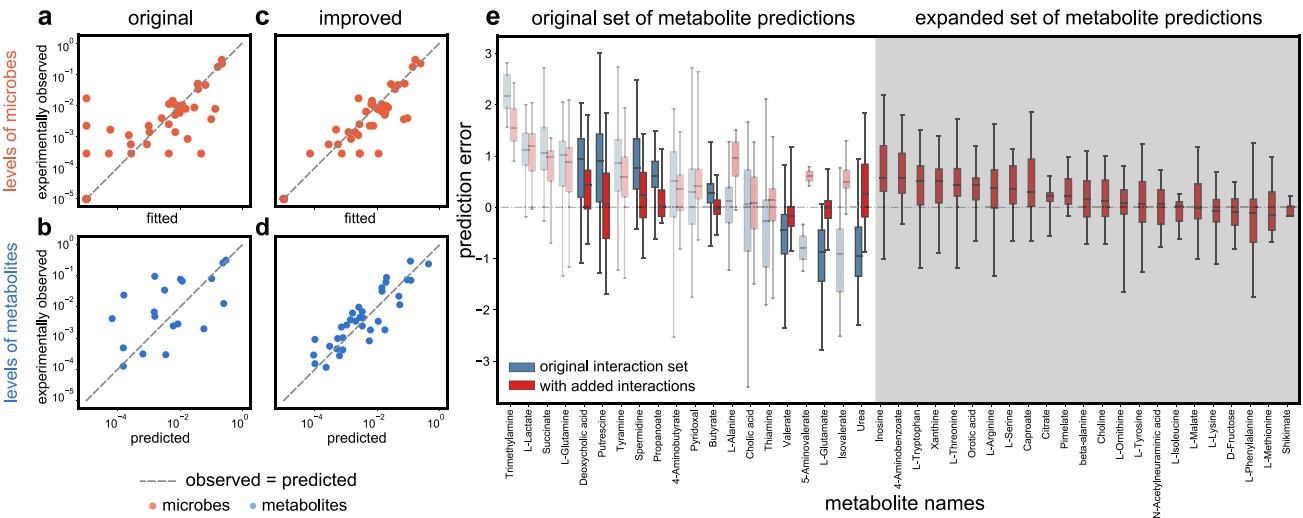

**Fig. 4 The effects of GutCP's predicted interactions on the gut microbiome and metabolome. a–d** Each panel compares the levels of microbial species (**a** and **c**; blue) or metabolites (**b** and **d**; orange) predicted by our ecological consumer-resource model ($x$ axis) with the experimentally measured levels ($y$ axis); the closer a point is to the marked line (indicating an exactly correct prediction), the better our predictive power. The Pearson correlation coefficients for panels (**a**) through (**d**) are as follows: (**a**) correlation 0.88, $P < 10^{-6}$, (**b**) 0.75, $P < 10^{-3}$, (**c**) 0.88, $P < 10^{-6}$, and (**d**) 0.77, $P < 10^{-6}$. All $P$ values are estimated using the two-sided $t$ test. The predictions using the original, known-set cross-feeding interactions (production and consumption links) are on the left, and using the additional cross-feeding interactions predicted by GutCP are on the right. **e** Box plot showing the improvement in prediction error of each metabolite in the fecal metabolome ($n = 41$ independent samples). In all boxplots, the middle line is the median, the lower and upper hinges correspond to the first and third quartiles, the upper whisker ranges from the hinge to the value $1.5 \times IQR$ (where IQR is the interquartile range) above the hinge and the lower whisker extends from the hinge to the value $1.5 \times IQR$ below the hinge, while all data points failing beyond the range of whiskers are plotted individually. Predictions errors using the original cross-feeding network are in blue, and those with added interactions predicted by GutCP are in red. Central bars indicate median, boxes and whiskers indicate quartiles. Metabolites for which GutCP improved predictions highly are shown in solid bold colors for illustration; those with faded colors represent modest improvements. The shaded gray part of the plot shows new metabolites whose levels GutCP helped predict, but the original cross-feeding network could not.

*difficile*; Fig. 3) increased the abundance of microbes producing 5-aminovalerate. These microbes then produced more 5-aminovalerate such that it was no longer underpredicted. Note that interactions such as these can only be inferred by causal and mechanistic models; this is because they alone can find such emergent, indirect effects of the microbiome on the metabolome.

**Validating the predicted interactions using evidence from genome sequences**. The full set of the interactions we predicted here (293) is quite large, which is why we provide them as a resource to guide experimental efforts in building a more complete list of cross-feeding interactions. While the experimental verification of our predictions is outside the scope of this study, we provide evidence suggesting that our predicted interactions are indeed consistent with the evidence from genome-scale metabolic networks[28,29,32], which annotates metabolic capabilities directly from genome sequences, but vastly overestimates the number of cross-feeding interactions. That is, if we use all the interactions predicted by genome-scale methods, we get a much poorer prediction accuracy for the metabolome profiles (average correlation coefficient 0.26 versus 0.62 using only the ground-truth interactions; Supplementary Fig. 6). This might be because genome-scale methods find all potential consumption and production links that the species are capable of, while only a fraction of them might be ecologically relevant and active in most gut microbiomes.

Nevertheless, we used these interactions as genomic evidence for validating GutCP's predictions. To do so, we calculated the fraction of predicted interactions that were also predicted by sequence-based methods (see "Methods" for details). As a control, we asked: if we picked interactions randomly from the set of all combinatorially possible interactions, what fraction of GutCP's predictions would still be present in the genome-based predictions? We found that 65% of our predicted interactions were also predicted by genome-based predictions, much higher than expected by chance (controls had ~20%; binomial test, $P < 10^{-6}$). This strongly suggests that GutCP's predicted interactions not only have ecological relevance but are also consistent with genome annotation results.

## Discussion

Inferring ecological interactions is crucial to building a mechanistic understanding of microbial communities as well as microbiomes[44]. To date, studies that have attempted this have focused on inferring species–species interactions[45–47]. Although knowledge of species–species interactions can be used to predict the possibility of coexistence between microbial species, the interactions themselves are dynamic and depend on environmental conditions[48,49]. This makes them difficult not only to verify but also to make subsequent predictions. Here, we have taken an alternative, but more powerful and mechanistic approach: that of inferring species–metabolite interactions (or cross-feeding interactions), which (1) subsume interspecies interactions, (2) depend more weakly on environmental conditions than species–species interactions, and (3) are simpler to experimentally verify. The new cross-feeding interactions predicted in this paper are a direct reflection of the metabolic capabilities of different microbial species and are thus easier to test through experiments. Our approach is grounded in a mechanistic model of the gut microbiome[20], which allows reliable causal inference between the metagenome and metabolome, compared with alternatives that depend merely on correlations between microbes and metabolites[11,50–52].

Using our algorithm, GutCP, we have provided here an atlas of 293 high-consensus cross-feeding interactions between 72

prominent gut microbial species and 221 gut metabolites. Given the general and broad applicability of GutCP, we anticipate that access to a larger number of experimental measurements of the gut metagenome and metabolome will help complete the inference of all relevant ecological metabolite-driven interactions in the microbiome. This is because GutCP helps to narrow down and pinpoint those interactions that are most likely to be present, and this is crucial because the number of possible cross-feeding interactions in the gut is very large (~30,000; see "Methods")[29,32]. Sampling all combinatorially possible interactions requires high-throughput experimental tests, far beyond the scope of what is currently possible. Further, genome-based metabolic network reconstruction methods are noisy and tend to predict more than 10,000 total interactions[29,32], tens of times more than the known ecologically relevant number of interactions in the gut[9]. With the proof-of-concept dataset that we used here, GutCP was able to narrow down this list from 30,000 to about 300, resulting in a 100-fold reduction of the required experimental throughput. While this is still a large number of experimental tests to perform, the complete table of predictions should serve as a resource guiding future experimentally tractable ecological inference in the gut microbiome.

We recognize that the ground-truth interaction network[9] we used in this study is an imperfect dataset, and likely to have a few biases and errors. To test this, we simulated a version of GutCP capable of not only adding new interactions but also removing known interactions. When we allowed GutCP to both add and remove interactions, model performance did not improve beyond that achieved purely by adding interactions (average correlation 0.75 when adding and removing, versus 0.75 when purely adding; Supplementary Fig. 7). This is why, for simplicity, GutCP only attempts to add new interactions instead of also attempting to remove old ones.

GutCP is conceptually analogous to gap-filling during flux-balance analysis (FBA) but operates at a community level[28,29,32]. Gap-filling infers intracellular metabolic reactions required for the growth of a single microbial species in a particular medium. GutCP infers extracellular, cross-feeding interactions required to better predict the levels of several microbial species and metabolites simultaneously. Thus, one can think of GutCP as a community-level gap-filling: where each microbial species is effectively a net chemical reaction, and new cross-feeding interactions add new links between species.

A key limitation of our method is that GutCP is built to reliably predict only those interactions which have a large and measurable impact on many samples of the gut microbiome and fecal metabolome, and is not likely to detect individual-specific interactions. Thus, it can miss interactions between rare species and metabolites, which might be real but not have a significant impact on metabolome predictions (GutCP works best when the improvement in predictions is of at least one order of magnitude). However, because GutCP prioritizes the discovery of species–metabolite interactions that systematically improve predictions across many samples, it is likely to detect those interactions that are less sensitive to environmental conditions, while ignoring others that are specific to certain conditions. Thus, the interactions predicted by GutCP are quite likely to be generally relevant in the gut microbiome.

GutCP also stands in contrast with previous correlation-based studies to infer microbe-microbe[53–57] and microbe–metabolite associations[11,50–52]. While these approaches are model-free and easy to compute, they lack any mechanistic understanding of the microbiome, and can thus cannot distinguish between direct and indirect effects of metabolites on microbes. Because of its explicit mechanistic and ecology-guided approach, GutCP can more naturally tell which microbe–metabolite interactions indicate a

direct versus an indirect association (see the examples in "Results"). Collectively, this work advances the field of integrative multi-omics, by suggesting a new way to integrate two -omics measurements (metagenomics and metabolomics) through causation, not merely correlation.

## Methods

**Datasets.** Throughout this study, we used a previously published dataset of simultaneous gut metagenome and fecal metabolome measurements from 41 human individuals[38]; this dataset was used as a proof-of-concept and was identical to the dataset used to calibrate the ecological consumer-resource model of the gut microbiome in this study (see Wang et al.[20] for the complete description of the model and how we processed the dataset). Briefly, the dataset measured 16S rRNA OTU abundances for gut metagenome measurements and CE-TOF mass spectrometry for quantitative fecal metabolome profile measurements. For the original, known set of cross-feeding interactions, we used a previously published database of experimentally verified and manually curated cross-feeding interactions, created specially for human gut microbiome studies[9]. We mapped the species in this database to the species in our experimental dataset as described previously in Wang et al.[20]. To compare our predicted interactions with genome-scale metabolic networks, we obtained semi-automatically reconstructed genome-scale metabolic models from Garza et al.[29]; this dataset had over 1500 genome-scale metabolic models, but we only used those that mapped to the 72 species and 221 metabolites in our dataset. While we did not explicitly include the host in our model (for simplicity), we included the number of trophic levels which were consistent with gut -omics data. These trophic levels are in part determined by the length of the host's gut[20].

**GutCP algorithm.** GutCP uses both a previously published ecological consumer-resource model and machine-learning optimization techniques. The ecological model we used in this paper was a previously published model that we developed, namely a trophic model of the human gut microbiome[20]. Our trophic model follows the discrete and stepwise flow of metabolite consumption and subsequent byproduct generation by microbial species in the gut. By knowing which species consume and produce which metabolites, this model can predict the fecal metabolome with high accuracy. Originally, we used the set of consumption and production abilities of each microbial species from a manually curated database, as described above. GutCP assumes that we can discover, infer and predict new cross-feeding interactions in the gut that are not present in the manually curated database by identifying that set of new interactions that further improve our estimate of the fecal metabolome. GutCP proceeds in discrete time steps, where each step resembles a Markov Chain Monte Carlo (MCMC) optimization method[34], but with a few key differences. GutCP consists of five major steps, detailed as follows.

Step 1: Setup, and measuring systematic biases. We start with an initial cross-feeding network, derived from the manually curated database of interactions in the gut microbiome. Each node in this bipartite network represented either a species or a metabolite. A directed link from a metabolite to a species denoted the ability of the species to consume the metabolite and was termed a consumption link. Similarly, a link from a species to a metabolite indicated its ability to secrete the metabolite and was termed a production link. While the network in the dataset was interconnected and not hierarchical, we showed in a previous study[20] that it could be roughly partitioned into four trophic levels, with the consumption of polysaccharides occurring at the top level, and the subsequent secretion of metabolic byproducts at later levels. We use our consumer-resource model with this original network on our dataset, and generate a set of metabolome estimates. We then calculate a systematic bias, $b_i$, for each metabolite and microbe predicted by the model, namely the difference between the predicted and experimentally measured levels, averaged over all samples in the dataset, as follows:

$$b_i = \frac{1}{N_s} \sum_{\alpha=1}^{N_s} (\log_{10}(p_{\alpha,i}) - \log_{10}(m_{\alpha,i})), \quad (1)$$

where $p_{\alpha,i}$ and $m_{\alpha,i}$ represent the predicted levels and experimentally measured levels, respectively, for sample $\alpha$ and microbe or metabolite, $i$. $N_s = 41$ is the number of samples in the dataset. We measure bias in logarithmic units to estimate the average order of magnitude of the bias. A large, positive bias indicates a systematic over-prediction, and a large, negative bias, a systematic under-prediction.

Step 2: Calculating priors and proposing a new link. GutCP then uses the initial systematic bias measurements to calculate the likelihood of missing links for a particular metabolite or microbial species. It assigns this likelihood by considering the magnitude and sign of the systematic bias for each microbe and metabolite. Specifically, it assigns the probability $\mathcal{P}_{i,j}^{con}$, that species $i$ consumes metabolite $j$, if species $i$ is underpredicted and/or if metabolite $j$ is overpredicted, as follows:

$$\mathcal{P}_{i,j}^{con} \propto e^{-3 \cdot (b_i - b_j)} + \kappa, \quad (2)$$

where $b_i$ and $b_j$ are the systematic biases of species $si$ and metabolite $j$ measured using Eq. (1), and $\kappa = 0.1$ is an arbitrarily chosen constant to ensure the addition of indirect cross-feeding interactions that do not depend on the levels of $i$ and $j$

specifically. Similarly, GutCP assigns the probability $\mathcal{P}_{i,j}^{\text{pro}}$, that species $i$ produces metabolite $j$, if metabolite $j$ is underpredicted, as follows:

$$\mathcal{P}_{i,j}^{\text{pro}} \propto e^{-3 \cdot b_j} + \kappa, \qquad (3)$$

where the symbols have the same meaning as in Eqs. (1) and (2). All associated prior probabilities on new links, $\mathcal{P}^{\text{con}}$ and $\mathcal{P}^{\text{pro}}$, are normalized to sum up to 1. GutCP then proposes the addition of a new link to the current cross-feeding network (originally, the given network) by choosing one link randomly using this prior probability distribution. Note that interactions are sampled randomly from the complete set of all 31,824 combinatorially possible interactions between the species and metabolites in the dataset (this estimate comes from multiplying the total number of unique species in the microbiome ($S = 72$ in our dataset), the total number of unique metabolites ($M = 221$ in our dataset) and the number 2 (for two possible types of interactions: production and consumption)). This is in contrast with interactions sampled only from genome-scale predictions, which we avoid for simplicity, and to reduce any biases due to genome annotation.

this estimate comes from multiplying the total number of unique species in the microbiome ($S = 72$ in our dataset), the total number of unique metabolites ($M = 221$ in our dataset) and the number 2 (for two possible types of interactions: production and consumption); this results in 31,824 possible interactions)

Step 3: Evaluating objective function with the proposed link. GutCP re-calculates the systematic bias for each metabolite and microbe predicted by our consumer-resource model, this time using the cross-feeding network with the newly proposed link. It then incorporates it into an objective function, $E$, defined as follows:

$$E = \frac{1}{N_s} \frac{1}{\mathcal{M}} \sum_{\alpha=1}^{N_s} \sum_{i=1}^{\mathcal{M}} |\log_{10}(p_{\alpha,i}) - \log_{10}(m_{\alpha,i})| \\ + \lambda_{\text{reg}} \cdot \mathcal{N}_{\text{added}} - \lambda_{\text{reward}} \cdot \mathcal{M}, \qquad (4)$$

where $\mathcal{M}$ is the number of metabolites predicted by the model that overlap with the experimentally measured metabolomes, and $\mathcal{N}_{\text{added}}$ is the total number of links added by GutCP. $\lambda_{\text{reg}}$ is a hyperparameter that penalizes the addition of new links by a fixed amount, and $\lambda_{\text{reward}}$ is a hyperparameter that encourages the algorithm to predict new metabolite levels that overlap with the experimentally measured metabolites. Specifically, we calculate $E$ both before and after the addition of the newly proposed link, and measure the difference between them, $\Delta E$.

Step 4: Accepting or rejecting the newly proposed link. GutCP accepts the newly proposed link with a probability proportional to the reduction in the value of the objective function, $\Delta E$. Essentially, GutCP accepts the link if $E$ reduces with a high probability, and accepts it if it increases with only a small probability; this is a common choice in such optimization algorithms, and in this case helps GutCP find links that combine with others later to together improve predictions as a pair. The probability of accepting a newly proposed link is $\mathcal{P}^{\text{accept}} \propto e^{\left(-\frac{\Delta E}{kT}\right)}$, where $\frac{1}{kT} = 5000$ is a calibrated effective energy, representing the effect of a randomly chosen link on the objective function.

Step 5: Stopping criteria. We then repeat steps 2–4 multiple times iteratively. GutCP stops when the change in the objective function $E$ due to carefully chosen links starts becoming comparable to changes due to a randomly added link. It does this by comparing the overall change in $E$ over the past 500 iterations. If this change is comparable to the change over 500 randomly chosen steps, GutCP stops.

**Calibration of hyperparameters.** To optimize the performance of GutCP's link discovery procedure, we calibrated the two hyperparameters in the objective function in Eq. (4), namely $\lambda_{\text{reg}}$ and $\lambda_{\text{reward}}$. For this, we chose a large range of these hyperparameters, between $10^{-4}$ and $10^{-2}$ for $\lambda_{\text{reg}}$, and $10^{-4}$ and $10^{-1}$ for $\lambda_{\text{reward}}$, each in multiples of 10. For each pair of hyperparameter values in this range, we ran GutCP and assessed its average performance at the end of 100 runs, where we used the same three measures of performance as throughout the text: (1) the correlation between the predicted and experimentally measured metabolome, (2) the log error (see main text), and (3) the number of metabolites in the measured metabolome predicted by our ecological consumer-resource model (Supplementary Figs. 4 and S5). We chose those values of the hyperparameters that simultaneously achieved the best combination of performances on all three measures. We finally chose the values $\lambda_{\text{reg}} = 10^{-3}$ and $\lambda_{\text{reward}} = 10^{-3}$ and used them for the results shown in the rest of this manuscript. In addition, there was no correlation between the performance of our model and the number of species in a sample (Supplementary Fig. 8), suggesting that there was no systematic effect of species diversity that we needed to account for.

**Obtaining the consensus-based atlas of predicted cross-feeding interactions.** To calculate a consensus-based set of cross-feeding predictions, we performed 100 independent runs of GutCP. For every link predicted over all the 100 runs, we measured its prevalence, that is, the fraction of runs in which GutCP discovered the link. To determine which links were inferred by GutCP more often than expected purely by chance, we also calculated a null distribution, which was equivalent to a binomial distribution; in the null, the probability of a link being discovered by chance was the average number of links discovered in any individual run (~140), divided by the total number of discoverable links. We used the null distribution to assign a $P$ value to each discovered link, and assigned those links with $P < 10^{-3}$ as

part of our consensus-based set of cross-feeding predictions (Supplementary Table 1). Increasing or decreasing the $P$ value threshold within the order of magnitude did not change the number of consensus predictions by >5%.

**Validating the predicted interactions using genome-scale metabolic models.** To validate the set of interactions predicted by GutCP, we used genome-scale metabolic models, which make predictions about metabolic reactions from genome sequences but are known to overestimate the number of metabolic reactions between species and metabolites in the environment. We used the dataset from Garza et al.[29], which contained over 1500 genome-scale metabolic models (GSMMs). We extracted only those models which were relevant to the 72 microbial species in our dataset. From each GSMM, we specifically extracted those reactions that were marked as extracellular, since those represented the consumption and production links that we are interested in (Supplementary Table 2). The extracted reactions also contained metabolites for which our dataset was missing experimental measurements, and we removed such reactions from our analysis. After extraction, we obtained a full list of all genome-based cross-feeding interactions relevant to the species and metabolites of interest (7381 predicted interactions). This list was a subset of all combinatorially possible interactions between these species and metabolites (31,824 total interactions); note that GutCP used the latter to propose new interactions (293 predicted interactions). Finally, to assess the overlap between GutCP's predictions and the genome-scale predictions, we measured the fraction of cross-feeding interactions predicted by GutCP that were presented in this list of GSMM-based predictions. Note that while making predictions, we did not provide the genome-scale predictions as an input to GutCP.

**Statistics.** To calculate correlation coefficients throughout the study, we used Pearson's correlation coefficient. Wherever we used $P$ values, we explained in "Methods" how we calculated them, since, for all such measurements in the study, we calculated the associated null distributions from scratch. All statistical tests were performed using standard numerical and scientific computing libraries in the Python programming language (version 3.5.2) and Jupyter Notebook (version 6.1).

**Reporting summary.** Further information on research design is available in the Nature Research Reporting Summary linked to this article.

## Data availability
There are no raw data associated with this study. Data related to the ground-truth network (ref. [9]) were extracted from https://www.nature.com/articles/ncomms15393, the processed data related to the metagenomes and metabolomes in the study are available at https://github.com/maslov-group/ML_human_gut/tree/master/data.

## Code availability
The code for both our simulations and statistical analysis, and for the GutCP algorithm, can be downloaded from https://github.com/maslov-group/ML_human_gut.

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

## Acknowledgements

We thank Ananthan Nambiar for help with interpreting evidence from genome sequences. A.G. is supported by the Gordon and Betty Moore Foundation as a Physics of Living Systems Fellow through grant number GBMF4513.

## Author contributions

S.M. designed the research and supervised the study; T.W. and A.G. performed simulations and calculations. V.D. performed data curation. All authors devised the study and wrote the paper.

## Competing interests

The authors declare no competing interests.
