## [Peer Review File · Nature Communications]

Reviewers' Comments:

Reviewer #1:

Remarks to the Author:

In their manuscript "Ecology-guided prediction of cross-feeding interactions in the human gut microbiome", Goyal et al use a computational model to predict the uptake of secretion of metabolites by different species in the human gut microbiome. With the ongoing growth in microbial genomic and metagenomic data, the next frontier is using these data to build models to understand and predict the factors shaping the human gut microbiome and this study is a valuable contribution. In this light, GutCP breaks important new ground in this field and the method may be expected to have significant impact. Please find below some comments and questions that should be addressed.

1. The problem that is addressed by GutCP is that FBA models derived from the genome sequence (GSMMs) predict too many transport reactions for a given bacterium and it is unlikely that these are all really present. On page 6, the authors state that automated genome annotation "vastly overestimates the number of cross-feeding interactions" but it is unclear what this statement is based on as there is no reference and no further details/evidence is given. To frame the challenge that is addressed by GutCP, it should be explained why this is a problem and why it arises. This should be explained at the start of the paper, not at the end - it is now mentioned in the Discussion: "GutCP helps to narrow down and pinpoint those interactions that are most likely to be present, and this is crucial because the number of possible cross-feeding interactions in the gut is extremely large ($\sim 30,000$)^{29,41}; sampling all possible interactions requires extremely high-throughput experimental tests, far beyond the scope of what is currently possible. Further, genome-based metabolic network reconstruction methods are noisy, and tend to predict more than 10,000 total interactions^{29,41}, tens of times more than the ecologically relevant number of interactions in the gut⁹." (A minor point to be addressed: I could not find the number 30,000 in reference 29 and 41.)

2. GutCP starts with a "ground truth dataset" of uptake/secretion reactions from ref #9 and adds possible links that are accepted if they improve the model fit. An obvious question that should be addressed is the performance of the optimal model compared to a situation where ALL the exchange reactions are added that are predicted from the genome? Showing this number and explaining why it does not work or why it is not so simple, would give the reader a better idea of the value of the optimization-based approach taken by GutCP versus a more inclusive approach.

3. In the same line of thought as the previous comment, it would be very interesting to show results of a run where reactions are not just added, but also removed from the model. This would allow GutCP to let go of the "ground truth dataset" - which is arguably also a biased and imperfect dataset - and start with either all exchange reactions or a random set of exchange reactions and, by removing reactions still reach an optimal cross-feeding network.

4. The main point that should be addressed is to include an independent benchmark. Currently, the authors present two lines of evidence supporting the performance of GutCP. First, they show that, by analyzing 3/4 of the data from the Sung et al paper (ref #56), GutCP can discover exchange reactions that are predicted from the remaining 1/4. This is not acceptable since those reactions are predicted by the same program and may be expected to have the same biases. Second, they show that the reactions that are added by GutCP are supported by encoded genes in the bacterial genome sequences of the community members. I'm not sure whether I got this right, but if the added reactions are sampled from the possible exchange reactions which are defined by the metabolic universe present in the FBA models/GSMMs (which in turn are based on the genome sequences too) then this is arguably biased as well. (I'm not sure because it is not very clear how the new exchange reactions are proposed - please clarify this.) Either way, although predicted exchange reactions are more often present in the corresponding GSMMs than expected by chance, it is still not an independent validation since one could have started from these exchange reactions

and there is likely a large overlap between these reactions and the "ground truth" set. What is needed is a truly independent benchmark, for example by leaving out half the "ground truth" dataset and observing how much of the second half can be recovered. Moreover the prediction quality could also be further assessed on a completely independent dataset such the one available from IBD patients (<https://www.nature.com/articles/s41467-019-10927-1#MOESM1>).

5. It is perhaps also not too surprising that "These new metabolites were indeed part of the experimentally measured metabolomes for these samples", since I guess that is why they were included in the sampling pool. More importantly, experimentally measured metabolites are even up-weighted by the algorithm, as explained in the Methods: "lambdareward is a hyper-parameter that encourages the algorithm to predict new metabolite levels that overlap with the experimentally measured metabolites." In my opinion this cannot be used as an argument supporting the performance of GutCP - if you disagree please discuss this in more detail.

6. "During extraction, we chose only those reactions which involved metabolites for which we had experimental measurements in our dataset. Doing this gave us a full list of all genome-based cross-feeding interactions relevant to the species and metabolites of interest. We then measured the fraction of cross-feeding interactions predicted by GutCP that were presented in this list of GSMM-based predictions." - I don't understand how one could expect any negatives here?

7. The abstract states "we predict hundreds of new experimentally untested cross-feeding interactions" but strictly these links are reactions that are sampled from the exchange reactions present in the GSMMs. So they are not "new" and this suggestion should be removed from the manuscript.

8. The manuscript talks about "links" or "connections" but this is confusing and obscures what is really going on in the model since the "links" do not represent cross-feeding between pairs of microbes, but uptake or secretion reactions of metabolites by specific microbes, i.e. uptake/secretion or exchange reactions with the common metabolite pool. This leads to confusing and seemingly circular statements in the manuscript, like "links resulting in the most accurate and optimal connection would be the most likely candidates for new cross-feeding interactions". To solve this, I suggest to avoid the words "link" and "connection" and consistently replace all occurrences of these words by the words "uptake", "secretion", or perhaps "exchange reaction", as appropriate.

9. The authors present "an atlas of 293 high-consensus cross-feeding interactions" that are expected to be true. How about negatives? So cross-feeding reactions that are expected from the models but that are never predicted by GutCP and may this be false?

10. The Github repository that should contain the code is empty.

Minor points:

The authors mention that current estimates of the number of cross-feeding links between gut microbes "vastly overestimate relevant cross-feeding interactions (i.e., they are beyond complete)"³⁰⁻³². But it is unclear what the expected range is and what this estimated number is based on. This should be explained and any literature cited that address the expected number of cross-feeding links.

The authors justify using the Wang et al 2019 model "because of its context and performance: it is [...] best able to connect the experimentally measured species composition of the gut microbiome with its resulting metabolic environment". It is unclear what "best" is based on, how was this measured and how much better was it than competing tools?

The authors state that "using only the species abundance from the microbiome, we use the model

to quantitatively estimate the microbiome's species and metabolomic composition." Currently, if a reader wants to know how this works they need to read the Wang et al paper. It would be very helpful to spend a few sentences to give the reader a brief idea of how this method works and how it builds on previous tools. For example, one question that was in my mind is, how are the exchange reactions weighted, i.e. how is it determined which metabolites are consumed/secreted more than others? Similarly, I was wondering how abundances of the microbes could be predicted, e.g. in the sentence "Reasonably, GutCP added several new consumption links to them, allowing these species increased growth and accurately-predicted abundances." I think that the answers to these and other questions will become clear if the Wang et al model is explained in a few sentences.

The formula for the prediction error is $\log_{10} ((\text{pred-meas})/\text{measurement})$, but I think it should be $\log_{10} (\text{absolute}(\text{pred-meas})/\text{measurement})$

I think red and pink should be gray and red in the caption of Figure 2.

References to Fig. 3c on page 4 should be 2c.

Production of D-Fructose is mentioned but not shown in Fig. 3a.

"Fig. 3b also shows that the network of new interactions have 2 clear type of bacteria" - it should be explained how these "types" are defined/quantified.

"Bacteroides, Ruminococcus and Bifidobacteria are known byproduct producers in the gut microbiome..." - requires a reference.

"...and as expected, GutCP predicted more production links for species in these genera^{14,37-39}" - is "more" significant? How was this tested?

"Consumers, on the other hand (right of Fig. 3b), typically occupy the lower trophic levels" - I would expect consumers to be higher up in the food web.

"Even by visual inspection, one can see that the newly predicted interactions bring the points much closer to the line of correct predictions." - It would be good to quantify this and add a P value.

"the amino acid lysine was under-predicted" (page 5) - where is this shown?

"With only a subset of the inferred links, the levels of such metabolites still remained under- or over-predicted" (page 5) - where is this shown?

"5-Aminovalerate (Fig. 3)" (page 5) - there is no 5-Aminovalerate in Fig. 3 - maybe that is the point but this is not clear.

Figure 4e shows a "Box plot showing the improvement in prediction error of each metabolite in the fecal metabolome." Only a very small set is shown, please explain why.

Blue/red have a different meaning in Fig. 4e than in Fig. 4a-d. It would be very helpful to have a consistent color scheme in all the figures in the manuscript.

Fig. 4e caption mentions "diamonds" but I see no diamonds.

Fig. 4e caption "Metabolites for which GutCP improved predictions highly are shown in solid bold colors for illustration; those with faded colors represent modest improvements." - It is unclear what is the difference between "high" and "modest" improvements: how is this quantified?

"boosts the growth of microbes that produce it" (page 6) - it is unclear what "it" refers to.

"As a control, we asked: if our predicted interactions were essentially random, what fraction of GutCP's predictions would still be present in the genome-based predictions?" (page 6). It would be good to state how "essentially random" is defined.

The first sentence of the Discussion should be rephrased.

Note that "species-metabolite interactions (or cross-feeding interactions)" (Discussion, page 6) also "depend on environmental conditions"!

"Even though at first glance, GutCP appears similar to gap-filling during flux-balance analysis (FBA)" - please rephrase, I don't think anyone would think the approaches are similar.

Consider placing the Methods before Results, I found it difficult to follow the Results at first. This might also be resolved by addressing minor point 3 above.

"models that mapped to the 72 species and 221 metabolites in our dataset", and "we specifically extracted those reactions that were marked as extracellular, since those represented the consumption and production links that we are interested in" - please provide all such lists of reactions/metabolites in Supplementary Tables and refer to them.

"this model can predict the fecal metabolome with relatively high accuracy" - relative to what? Please specify/quantify.

"We start with an initial cross-feeding network" - defined how? How many trophic layers? More details are needed.

Reviewer #2:

Remarks to the Author:

This study proposes a computational approach that is claimed to be capable to identify novel and undetectable cross-feeding interactions within complex microbial ecosystem. Method has been explained in a well-organized and easy-to-understand manner, and reported results revealed potential of method to provide some mechanistic insight into microbial communities. However, there are some aspects that are needed to be explained more clearly or to be added to the study.

1. It has been mentioned that GutCP is generalizable and can be used with other mechanistic models, but authors preferred to use their own previously published consumer-resource model. To evaluate the power of the method, it would be helpful to employ at least one of the other mentioned models and present the results.

2. Metabolic interaction between host and microbiome is an important factor in shaping the microbial communities and inter-species interactions. How and where in the method they include this factor.

3. When adding new cross-feeding interactions, how they define the directionality of the new links?

4. Authors evaluated the results using the systematic bias: the average deviation of the predictions from measured levels across all samples. Samples usually are collected in different conditions and species show different metabolic profiles in different conditions. How model solve the effect of outlier samples and changed metabolic profiles?

5. As the complex systems current and future states are sensitive to initial conditions, how model decides which new links to add first? And how it guarantees that starting from different links and following different trajectories will not result in biased predictions? How model tackle with risk of trapping in local optimum solutions and missing global optimum?

6. Authors used test dataset from 41 human individuals, comprising 221 metabolites and 72 microbial species. How performance of the algorithm evaluated for larger communities consisting of higher number of species? especially knowing that gut microbiota consists of thousands of species and metabolites. It is required to perform sensitivity analysis based on size of the microbial community.

7. Authors found that the newly predicted interactions had both direct and indirect effects on metabolite levels. Implementing indirect effect is not explained well. How it has been calculated and what will be the consequence of biased indirect effect on predictions.

8. Genome scale metabolic models (GEMs) have been used to validate the predictions, although authors earlier mentioned that GEMs are over estimating interaction. How they control over estimation? I recommend a validation on a new cohort or some in vitro experiments.

Reviewer #1 (Remarks to the Author):

In their manuscript "Ecology-guided prediction of cross-feeding interactions in the human gut microbiome", Goyal et al use a computational model to predict the uptake of secretion of metabolites by different species in the human gut microbiome. With the ongoing growth in microbial genomic and metagenomic data, the next frontier is using these data to build models to understand and predict the factors shaping the human gut microbiome and this study is a valuable contribution. In this light, GutCP breaks important new ground in this field and the method may be expected to have significant impact. Please find below some comments and questions that should be addressed.

We are grateful to the reviewer for this positive evaluation. We are particularly thankful for the detailed comments, suggestions and corrections that the referee offered, which we believe helped greatly improve the quality and clarity of our manuscript.

1. The problem that is addressed by GutCP is that FBA models derived from the genome sequence (GSMMs) predict too many transport reactions for a given bacterium and it is unlikely that these are all really present. On page 6, the authors state that automated genome annotation "vastly overestimates the number of cross-feeding interactions" but it is unclear what this statement is based on as there is no reference and no further details/evidence is given. To frame the challenge that is addressed by GutCP, it should be explained why this is a problem and why it arises. This should be explained at the start of the paper, not at the end - it is now mentioned in the Discussion: "GutCP helps to narrow down and pinpoint those interactions that are most likely to be present, and this is crucial because the number of possible cross-feeding interactions in the gut is extremely large (~ 30,000)29,41; sampling all possible interactions requires extremely high-throughput experimental tests, far beyond the scope of what is currently possible. Further, genome-based metabolic network reconstruction methods are noisy, and tend to predict more than 10,000 total interactions 29,41, tens of times more than the ecologically relevant number of interactions in the gut9."

We whole-heartedly agree that clarifying why genome-scale models are insufficient will help better motivate the problem addressed by GutCP.

The key arguments are two-fold: (1) since genome-scale models are based on genome annotations, they comprise both active and inactive interactions in the gut; and (2) in total, they predict >10,000 interactions, typically ~30% of all combinatorially possible interactions, and using all of these interactions leads to less accurate predictions of the fecal metabolome compared with both experimentally verified interactions, as well as with the interactions predicted by GutCP (Fig. S6; see also response to reviewer #1, major point #2).

We have now added these arguments, as well as the appropriate references, to the Introduction.

Lines 29-35: *"Indirect methods, which chiefly comprise inferring the metabolic activity of gut microbes from their genome sequences, are noisy, lack curation and vastly overestimate relevant cross-feeding interactions (i.e., they are "beyond complete"; since they are based on genome annotations, they comprise both active and inactive interactions in the gut)."*

(A minor point to be addressed: I could not find the number 30,000 in reference 29 and 41.)

We estimated nearly 30,000 combinatorially possible interactions using the number of species and metabolites in our dataset. References 29 and 41 were instead used to estimate the number of interactions predicted by genome-scale models. The possible number of interactions can be calculated by multiplying the total number of unique species in the microbiome ($S = 72$ in our dataset), the total number of unique metabolites ($M = 221$ in our dataset) and the number 2 (for 2 possible types of interactions: production and consumption); this gives us an estimate of 31,824 interactions, which we approximated as 30,000.

We have added this clarification in the Methods section where we discuss this estimate.

Lines 606-615: *“Note that interactions are sampled randomly from the complete set of all 31,824 combinatorially possible interactions between the species and metabolites in the dataset (this estimate comes from multiplying the total number of unique species in the microbiome ($S = 72$ in our dataset), the total number of unique metabolites ($M = 221$ in our dataset) and the number 2 (for 2 possible types of interactions: production and consumption)).”*

2. GutCP starts with a "ground truth dataset" of uptake/secretion reactions from ref #9 and adds possible links that are accepted if they improve the model fit. An obvious question that should be addressed is the performance of the optimal model compared to a situation where ALL the exchange reactions are added that are predicted from the genome? Showing this number and explaining why it does not work or why it is not so simple, would give the reader a better idea of the value of the optimization-based approach taken by GutCP versus a more inclusive approach.

This is an excellent suggestion. We performed new simulations to compare the performance of our model when we use all the exchange reactions predicted using genome sequences with that of GutCP. Indeed, we found that using all exchange reactions resulted in significantly lower performance (average Pearson correlation 0.26 compared with 0.74 for the optimal GutCP model).

We have now added new text in the Results section, as well as a new supplementary figure (Fig. S6), to highlight the value of optimizing which interactions to add (taken by GutCP), as opposed to adding all predicted interactions (from genome sequences).

Lines 386-394: *“That is, if we use all the interactions predicted by genome-scale methods, we get a much poorer prediction accuracy for the metabolome profiles (average correlation coefficient 0.26 versus 0.62 using only the ground-truth interactions; Fig. S6). This might be because genome-scale methods find all potential consumption and production links that the species are capable of, while only a fraction of them might be ecologically relevant and active in most gut microbiomes.”*

3. In the same line of thought as the previous comment, it would be very interesting to show results of a run where reactions are not just added, but also removed from the model. This would allow GutCP to let go of the "ground truth dataset" - which is arguably also a biased and imperfect dataset - and start with either all exchange reactions or a random set of exchange reactions and, by removing reactions still reach an optimal cross-feeding network.

We thank the reviewer for pointing this out and giving us a valuable opportunity to explain why GutCP does not remove reactions. Indeed, we had tried a version of GutCP which both added and removed reactions. We found that doing this did not improve the performance of the model beyond that achieved

purely by adding reactions (average correlation 0.75 when adding and removing, versus 0.75 when purely adding; Fig. S7).

We suspect that this is because GutCP can only reliably add or remove those reactions that significantly impact the performance of our model. Thus, removing reactions in the ground truth dataset that did not significantly affect the model's predictions will not tend to greatly improve or reduce its performance.

Further, because complex microbial communities such as the human gut microbiome have been shown to contain a lot of metabolic redundancies, we refrained from removing reactions which might otherwise be true, but not manifest as improvements to metabolome predictions.

In light of these results and arguments, we chose a more parsimonious version of GutCP which only adds (and doesn't remove) reactions. The reviewer's comment made us realize that these results might still be of general interest, and so we now include them in a supplementary figure (Fig. S7) and discuss them in the text.

Lines 462-473: *"We recognize that the ground truth dataset we used in this study is an imperfect dataset, and likely to have a few biases and errors. To test this, we simulated a version of GutCP capable of not only adding new interactions, but also removing known interactions. When we allowed GutCP to both add and remove interactions, model performance did not improve beyond that achieved purely by adding interactions (average correlation 0.75 when adding and removing, versus 0.75 when purely adding; Fig. S7). This is why, for simplicity, GutCP only attempts to add new interactions instead of also attempting to remove old ones."*

4. The main point that should be addressed is to include an independent benchmark. Currently, the authors present two lines of evidence supporting the performance of GutCP.

We understand the reviewer's concern regarding possible biases in our evidence supporting the performance of GutCP, and address it below.

First, they show that, by analyzing 3/4 of the data from the Sung et al paper (ref #56), GutCP can discover exchange reactions that are predicted from the remaining 1/4. This is not acceptable since those reactions are predicted by the same program and may be expected to have the same biases.

The reviewer is indeed correct that predicting interactions from two separate parts of the same dataset and comparing them to each other might be inherently biased. We wish to clarify that this is not what we did in our study, and apologize if our original text made it seem like we did.

Instead, we employed a standard technique in machine learning (cross-validation) by dividing our -omics data into two subsets: training (3/4 of the data) and test (1/4 of the data) subsets. We then employed GutCP on the training set to discover the new links, added them to the ground-truth interactions of [Sung et al]; finally, we evaluated the accuracy of metabolome predictions from the resulting network on the test set. Note that we did not discover exchange reactions from the test set, only from the training set.

We showed that the interactions predicted using the training subset of the data (3/4 of it) could also improve the accuracy of metabolome predictions in the unseen, test subset (the remaining 1/4 of it). This suggested that the discovered reactions were not specific to, in other words did not over fit, the training

data. This supports the idea that the interactions discovered by GutCP are in part generalizable to other datasets.

We recognize that the original text corresponding to this section of the manuscript might confuse readers, and have therefore re-written it to clarify how and why we performed cross-validation.

Lines 197-217: *“To test if the cross-feeding interactions predicted by GutCP are generalizable to unknown datasets, we performed 4-fold cross-validation. We used a sample -omics dataset of the gut microbiome and metabolome sampled from 41 human individuals, comprising 221 metabolites and 72 microbial species (data from ref. 38). We split our -omics dataset into two subsets: training (three-fourths of the individuals) and test (one-fourth of the individuals) subsets. We then ran GutCP on the training subset to discover new interactions, and added them to the ground-truth interactions taken from ref. 9. Doing so resulted in a network of cross-feeding interactions learnt only from the training subset of the data. Finally, we evaluated the improvement in accuracy of metabolome predictions resulting from the trained network on the unseen, test subset of the data. We repeated this process 3 times, each time splitting the full dataset into a training subset (with a randomly chosen three-fourths of the individuals) and test subset (with the remaining one-fourth of the individuals); finally, we calculated the average improvement in prediction accuracy over all 4 splits.”*

Second, they show that the reactions that are added by GutCP are supported by encoded genes in the bacterial genome sequences of the community members. I'm not sure whether I got this right, but if the added reactions are sampled from the possible exchange reactions which are defined by the metabolic universe present in the FBA models/GSMMs (which in turn are based on the genome sequences too) then this is arguably biased as well. (I'm not sure because it is not very clear how the new exchange reactions are proposed – please clarify this.)

We wish to clarify that the newly proposed interactions added by GutCP were **not** sampled from the reactions in the genome sequences, but instead they were probabilistically chosen from all combinatorially possible interactions between any species and metabolites present in our dataset. Therefore, for S species and M metabolites found in the dataset, there were a total of $2SM$ combinatorially possible interactions (the 2 appears because for each species-metabolite pair, there is one production and one consumption interaction). To reiterate, the links added by GutCP were not in any way biased towards links in the universe present in the FBA models/GSMMs.

Instead, we used the GSMMs to provide and serve as an independent genomic benchmark of GutCP's predicted interactions. We found that 65% of interactions predicted by GutCP were also detected in the genomes of the corresponding species, according to GSMMs (enrichment P value $< 10^{-6}$).

We thank the reviewer for highlighting that it was unclear how GutCP proposed new interactions, and what the GSMMs were used for. In the revised version, we have modified the text in several parts to make this clear for readers.

Lines 171-175: *“This new link is chosen randomly from the entire set of combinatorially possible links (see Methods; for \$\$\$ species, \$\$M\$ metabolites, and two kinds of links (consumption and production), there are a total of \$2SM\$ combinatorially possible links).”*

Either way, although predicted exchange reactions are more often present in the corresponding GSMMs than expected by chance, it is still not an independent validation since one could have started from

these exchange reactions and there is likely a large overlap between these reactions and the “ground truth” set. What is needed is a truly independent benchmark, for example by leaving out half the “ground truth” dataset and observing how much of the second half can be recovered. Moreover the prediction quality could also be further assessed on a completely independent dataset such the one available from IBD patients (<https://www.nature.com/articles/s41467-019-10927-1#MOESM1>).

We completely agree that a wholly independent benchmark dataset would be an ideal test, but we emphasize that such data are currently not available. The reviewer suggested using data from IBD patients, and this is in principle a good suggestion. However, the microbiomes and metabolomes of healthy human individuals can often be very different from those of diseased patients. Indeed the reference that we used for our ground truth dataset provides another network for patients with type-2 diabetes. This is why we do not believe that testing GutCP’s predicted interactions (which was done using healthy humans) on IBD patients is an appropriate test to provide the suggested benchmark.

We have independently been in communication with two groups, one in the US, and another in Israel, about access to new datasets which will provide such benchmarks, but since these data are unpublished, providing these tests remains outside the scope of this manuscript.

5. It is perhaps also not too surprising that “These new metabolites were indeed part of the experimentally measured metabolomes for these samples”, since I guess that is why they were included in the sampling pool. More importantly, experimentally measured metabolites are even up-weighted by the algorithm, as explained in the Methods: “ λ is a hyper-parameter that encourages the algorithm to predict new metabolite levels that overlap with the experimentally measured metabolites.” In my opinion this cannot be used as an argument supporting the performance of GutCP - if you disagree please discuss this in more detail.

We agree with the reviewer that the observation—that several newly predicted links metabolites were part of the experimentally measured set—is indeed not surprising. However, what we thought was somewhat surprising and supportive of GutCP was that: the new interactions or links for such metabolites resulted in quantitatively accurate predictions.

We have re-written this sentence in the text to avoid any potential confusion to readers.

Lines 309-314: *“As expected, these new metabolites were indeed part of the experimentally measured metabolomes for these samples. Encouragingly, GutCP could predict their levels with an accuracy comparable with the original set of metabolites (compare Fig. 4d with Fig. 4c).”*

6. “During extraction, we chose only those reactions which involved metabolites for which we had experimental measurements in our dataset. Doing this gave us a full list of all genome-based cross-feeding interactions relevant to the species and metabolites of interest. We then measured the fraction of cross-feeding interactions predicted by GutCP that were presented in this list of GSMM-based predictions.” - I don’t understand how one could expect any negatives here?

We understand the reviewer’s confusion due to our insufficient description and apologize for it. To clarify, here we used two different sets of interactions, as explained below.

The first is a set of all combinatorially possible interactions between the experimentally measured species and metabolites in our gut dataset (as mentioned in a previous response, there are $2SM$ such

interactions, where S is the number of species and M , the number of metabolites). This set of roughly 30,000 interactions is the one that GutCP uses to propose new interactions. Specifically, for the dataset we used, it accepted 293 of these combinatorially possible interactions.

The second is a set of interactions predicted using genome sequence-based methods for the species and metabolites in our gut microbiome and metabolome dataset. This is a small subset of all combinatorially possible interactions, and for the dataset we used, comprises 7,381 interactions. This set serves as data for genomic evidence of interactions.

In the text highlighted, we wished to ask the following question: what fraction of the 293 interactions that GutCP predicted were also part of the 7,381 interactions predicted by GSMMs? In other words, we wanted to ask if GutCP's predictions were enriched in the GSMM predictions. The answer was: 65% of these interactions were confirmed in the GSMM dataset.

It is indeed possible to get negatives here. For instance, when GutCP predicts interactions that are possible (part of the 30,000), but not part of the GSMM-based predictions (that is, not part of the 7,381). During the optimization procedure followed by GutCP, we do not distinguish between which interactions were predicted by GSMMs, and which not; this avoids any bias towards GSMM-based predictions.

We have re-written this part of the text so that this aspect of the method is clear to readers.

Lines 703-716: *“The extracted reactions also contained metabolites for which our dataset was missing experimental measurements, and we removed such reactions from our analysis. After extraction, we obtained a full list of all genome-based cross-feeding interactions relevant to the species and metabolites of interest (7,381 predicted interactions). This list was a subset of all combinatorially possible interactions between these species and metabolites (31,824 total interactions); note that GutCP used the latter to propose new interactions (293 predicted interactions). Finally, to assess the overlap between GutCP's predictions and the genome-scale predictions, we measured the fraction of cross-feeding interactions predicted by GutCP that were presented in this list of GSMM-based predictions. Note that while making predictions, we did not provide the genome-scale predictions as an input to GutCP.”*

7. The abstract states "we predict hundreds of new experimentally untested cross-feeding interactions" but strictly these links are reactions that are sampled from the exchange reactions present in the GSMMs. So they are not "new" and this suggestion should be removed from the manuscript.

We apologize that this was not clear. As mentioned in our response to point 6, we allow GutCP to add all combinatorially possible interactions (from a set of nearly 30,000). We do not constrain ourselves to only those interactions that were predicted by GSMMs. Instead, the GSMM-based interactions served as a completely independent test dataset, providing genomic evidence of the predicted interactions.

Moreover, 65% of the 293 predictions were not present in the GSMM-based predictions, and hence comprise truly new, previously undescribed interactions.

Since the predicted set was derived from a full set of all combinatorially possible interactions between the given species and metabolites, and since these interactions have not yet been directly tested, we believe that the word “new” is appropriate in this context.

8. The manuscript talks about "links" or "connections" but this is confusing and obscures what is really going on in the model since the "links" do not represent cross-feeding between pairs of microbes, but uptake or secretion reactions of metabolites by specific microbes, i.e. uptake/secretion or exchange reactions with the common metabolite pool. This leads to confusing and seemingly circular statements in the manuscript, like "links resulting in the most accurate and optimal connection would be the most likely candidates for new cross-feeding interactions". To solve this, I suggest to avoid the words "link" and "connection" and consistently replace all occurrences of these words by the words "uptake", "secretion", or perhaps "exchange reaction", as appropriate.

We thank the reviewer for spotting a potential source of confusion to readers, which was due to our oversight while writing the manuscript. We have taken three steps to address this comment: (1) we have replaced the word "interactions" with "cross-feeding reactions" in appropriate parts of the manuscript; (2) we explicitly define and explain what we mean by "links" early on in the manuscript (as either uptake or secretion reactions, forming the basis of cross-feeding) and continue to use it in a consistent manner; and finally (3) we have fixed all potentially confusing and circular-sounding statements such as the one highlighted by the reviewer in the text.

Lines 96-114: *"We hypothesized that adding new, yet-undiscovered cross-feeding interactions would improve our ability to predict the levels of metabolites with our mechanistic and causal model. Specifically, we predict that the set of undiscovered interactions resulting in the most accurate and optimal improvement in predictions would be the most likely candidates for true cross-feeding interactions. Inferring such an optimal set of new cross-feeding interactions is the main logic driving GutCP. In what follows, we sometimes refer to cross-feeding reactions (i.e. metabolite consumption or production by microbes) as "links" in an overall cross-feeding network of the gut microbiome, whose nodes are microbes and metabolites (Fig. 1a; metabolites in blue, microbes in orange); the links themselves are directed edges connecting the nodes. Links can be of two types: consumption or nutrient uptake reactions (from nutrients to microbes) and production or nutrient secretion reactions (from microbes to their metabolic byproducts)."*

9. The authors present "an atlas of 293 high-consensus cross-feeding interactions" that are expected to be true. How about negatives? So cross-feeding reactions that are expected from the models but that are never predicted by GutCP and may this be false?

It is indeed possible to have negatives, i.e., expected interactions that GutCP is unable to predict, but this depends in part on two things: (1) on the species and metabolites being talked about, and (2) whether the species and metabolites between which an interaction is expected has a measurable impact on the gut's metabolome profile. If the species and metabolites are those that were part of our dataset, and have an impact on the metabolome profile, we believe that negatives are possible but unlikely. In other cases, negatives can indeed occur with reasonable probabilities, and this is a caveat of our method.

We have added this caveat of GutCP to the Discussion section.

Lines 485-501: *"A key limitation of our method is that GutCP is built to reliably predict only those interactions which have a large and measurable impact on the gut microbiome and fecal metabolome profiles. Thus, it can miss interactions between rare species and metabolites, which might be real but not have a significant impact on metabolome predictions (GutCP works best when the improvement in predictions is of at least one order of magnitude). More upcoming data from controlled experiments in*

artificial gut communities, where such species and metabolites may be more abundant, might help GutCP detect such otherwise-overlooked interactions."

10. The Github repository that should contain the code is empty.

We are extremely sorry for this unintentional mistake. We have uploaded the code to the Github repository, and it can be accessed at the following URL (updated in the text):

https://github.com/maslov-group/ML_human_gut

Minor points:

The authors mention that current estimates of the number of cross-feeding links between gut microbes "vastly overestimate relevant cross-feeding interactions (i.e., they are beyond complete)"³⁰⁻³². But it is unclear what the expected range is and what this estimated number is based on. This should be explained and any literature cited that address the expected number of cross-feeding links.

We have addressed this point by explicitly mentioning the observed number of cross-feeding interactions according to genome-scale models, and providing appropriate references for them. While we could not find literature indicating the exact number of expected interactions, the large number of observed interactions drastically reduce our ability to make accurate metabolome predictions (Fig. S6).

The authors justify using the Wang et al 2019 model "because of its context and performance: it is [...] best able to connect the experimentally measured species composition of the gut microbiome with its resulting metabolic environment". It is unclear what "best" is based on, how was this measured and how much better was it than competing tools?

We apologize for the lack of clarity about our previous results. As reported in Wang et al (2019), our model outperformed the state of the art (at the time of publication) method to predict the gut's metabolome: namely Garza et al, *Nature Microbiology* (2018). Specifically, our model achieved a better Pearson correlation between the predicted and observed metabolome (0.62) compared with Garza et al's method (~0.5).

The authors state that "using only the species abundance from the microbiome, we use the model to quantitatively estimate the microbiome's species and metabolomic composition." Currently, if a reader wants to know how this works they need to read the Wang et al paper. It would be very helpful to spend a few sentences to give the reader a brief idea of how this method works and how it builds on previous tools. For example, one question that was in my mind is, how are the exchange reactions weighted, i.e. how is it determined which metabolites are consumed/secreted more than others? Similarly, I was wondering how abundances of the microbes could be predicted, e.g. in the sentence "Reasonably, GutCP added several new consumption links to them, allowing these species increased growth and accurately-predicted abundances." I think that the answers to these and other questions will become clear if the Wang et al model is explained in a few sentences.

We are grateful to the reviewer for this suggestion. We agree that it is helpful to familiarize readers with a brief idea of how our trophic model works, and have now added a few sentences to the first section of Results providing a brief explanation. The questions indicated by the reviewer helped guide our writing of these sentences, and we thank them for clearly articulating them.

Lines 122-150: *“For each sample, using only the species abundance from the microbiome, we use the model to quantitatively estimate the microbiome's species and metabolomic composition. Briefly, we assume that a defined set of polysaccharides, common to human diets, are available as the nutrient intake to the gut (nutrients 1 and 4 in Fig. 1a). We calculate the microbiome and metabolome profiles separately for each individual, which contain a different set of microbial species in their guts. At the first trophic level, all microbial species that are capable of using the polysaccharides (indicated by the pink arrows in Fig. 1a) consume each of them in proportion to their abundances (microbes a, b and c in Fig. 1a). They subsequently secrete a fixed fraction of the consumed nutrients as metabolic byproducts; every species at this trophic level secretes all the metabolic byproducts it is known to secrete (blue arrows in Fig. 1a) in equal proportion (nutrients 2--6 in Fig. 1a). At the next trophic level, all species detected in the individual's gut which can consume the newly secreted byproducts consume them as nutrients, secreting a new set of byproducts, and this continues for four trophic levels (not shown in Fig. 1a for simplicity). At the end of this process, all metabolites which remain unconsumed by the community comprise the metabolome of the individual, and the microbial species which consume nutrients and grow comprise the microbiome of the individual (for a complete description, see Methods and previous work).”*

The formula for the prediction error is $\log_{10} ((\text{pred-meas})/\text{measurement})$, but I think it should be $\log_{10} (\text{absolute}(\text{pred-meas})/\text{measurement})$

We used the following expression to measure log error: $\log_{10} [(\text{predicted}) / (\text{measurement})]$.

I think red and pink should be gray and red in the caption of Figure 2.

We thank the reviewer for pointing this out and have corrected it.

References to Fig. 3c on page 4 should be 2c. **Corrected.**

Production of D-Fructose is mentioned but not shown in Fig. 3a. **Corrected.**

"Fig. 3b also shows that the network of new interactions have 2 clear type of bacteria" - it should be explained how these "types" are defined/quantified.

We separated types based on the directionality of the links. We have now explained it in the text.

"Bacteroides, Ruminococcus and Bifidobacteria are known byproduct producers in the gut microbiome..." - requires a reference. **We have added the references.**

"...and as expected, GutCP predicted more production links for species in these genera^{14,37-39}" - is "more" significant? How was this tested?

Here, by "more", we meant numerically more links were added for species in these genera compared with other genera. Since each added link has already passed a statistical test for significance, we believe a numerical comparison is sufficient for this statement.

"Consumers, on the other hand (right of Fig. 3b), typically occupy the lower trophic levels" - I would expect consumers to be higher up in the food web.

We meant byproduct consumers, which we would expect to occupy lower trophic levels, due to the increased prevalence of byproducts at those levels. We have replaced “consumers” with the more descriptive “known byproduct consumers” in the appropriate location in the text.

"Even by visual inspection, one can see that the newly predicted interactions bring the points much closer to the line of correct predictions." - It would be good to quantify this and add a P value.

We have now added appropriate *P* values in the text to support this argument.

Fig. 4 caption, in red: “*The Pearson correlation coefficients for panels (a) through (d) are as follows: (a) correlation 0.88, $P < 10^{-6}$, (b) 0.75, $P < 10^{-3}$, (c) 0.88, $P < 10^{-6}$, and (d) 0.77, $P < 10^{-6}$.*”

"the amino acid lysine was under-predicted" (page 5) - where is this shown?

We apologize for this inadvertent error; indeed lysine was not predicted at all. We have removed mention of this from the revised text.

"With only a subset of the inferred links, the levels of such metabolites still remained under- or over-predicted" (page 5) - where is this shown?

We did not show this, only indicated to the reader that it was the case. We have now quantified this and added it to the sentence.

Lines 354-355: “*With only a subset of the inferred links, the levels of such metabolites still remained under- or over-predicted (on average, by 1 order of magnitude).*”

"5-Aminovalerate (Fig. 3)" (page 5) - there is no 5-Aminovalerate in Fig. 3 - maybe that is the point but this is not clear.

Indeed, our original explanation lacks clarity. We meant to say that sometimes, GutCP manages to improve predictions for metabolite levels even without adding any new interactions which produce or consume them. This is because interactions added to other species and metabolites percolate down the trophic levels and have effects on other metabolites and species. In this case, we took the example of 5-Aminovalerate, which was absent from Fig. 3 (because no interaction was predicted for it), but showed a prediction error improvement in Fig. 4e (because of several other interactions, such as the consumption of putrescine by *C. difficile*).

We have re-written this sentence to clarify this to future readers.

Lines 357-374: “*Indirect effects comprise any discovered links where GutCP improves the prediction for a metabolite without adding a link that produces or consumes it. The improvement in prediction comes entirely from other added links, which can increase or decrease the levels of microbes that produce (or consume) that metabolite. For example, GutCP inferred no new consumption or production links for 5-Aminovalerate (no predicted interactions in Fig. 3), but adding other links (e.g., the consumption of putrescine by *Clostridium difficile*; Fig. 3) increased the abundance of microbes producing 5-Aminovalerate. These microbes then produced more 5-Aminovalerate such that it was no longer under-predicted}. Note that interactions such as these can only be inferred by causal and mechanistic*

models; this is because they alone can find such emergent, indirect effects of the microbiome on the metabolome."

Figure 4e shows a "Box plot showing the improvement in prediction error of each metabolite in the fecal metabolome." Only a very small set is shown, please explain why.

In Fig. 4e, we included only those metabolites for which we had experimental measurements in order to calculate a prediction error. We did not show other metabolites whose levels we could predict, but were not part of the experimentally measured set; this is because we could not measure a prediction error for them.

Blue/red have a different meaning in Fig. 4e than in Fig. 4a-d. It would be very helpful to have a consistent color scheme in all the figures in the manuscript.

We agree and have changed the colors in the figures to have a more consistent color scheme throughout the text.

Fig. 4e caption mentions "diamonds" but I see no diamonds.

We apologize. We have removed the diamonds and their mention from the text.

Fig. 4e caption "Metabolites for which GutCP improved predictions highly are shown in solid bold colors for illustration; those with faded colors represent modest improvements." - It is unclear what is the difference between "high" and "modest" improvements: how is this quantified?

We did not quantify this. Instead, we used the following notion to distinguish between modest and high improvements. There were cases where GutCP's predictions reduced the error so that the average prediction error was no longer statistically different from 0 (originally, they were); we called such improvements "high"; all other improvements were classified as "modest".

"boosts the growth of microbes that produce it" (page 6) - it is unclear what "it" refers to.

"It" refers to 5-Aminovalerate. We have rephrased this sentence for clarity.

*Lines 363-370: "{For example, GutCP inferred no new consumption or production links for 5-Aminovalerate (Fig. 3), but adding other links (e.g., the consumption of putrescine by *Clostridium difficile*) increased the abundance of microbes producing 5-Aminovalerate. These microbes then produced more 5-Aminovalerate such that it was no longer under-predicted"*

"As a control, we asked: if our predicted interactions were essentially random, what fraction of GutCP's predictions would still be present in the genome-based predictions?" (page 6). It would be good to state how "essentially random" is defined.

We have re-written this sentence to improve clarity.

Lines 399-402: "As a control, we asked: if we picked interactions randomly from the set of all possible interactions, what fraction of GutCP's predictions would still be present in the genome-based predictions?"

The first sentence of the Discussion should be rephrased.

We have added a missing “as well as” to the sentence so that it is clear.

Lines 410-412: *“Inferring ecological interactions is crucial to building a mechanistic understanding of microbial communities as well as microbiomes.”*

Note that "species-metabolite interactions (or cross-feeding interactions)" (Discussion, page 6) also "depend on environmental conditions"!

We agree, and thank the reviewer for pointing this out. We have clarified this in the Discussion now, and added that we believe that species-metabolite interactions would depend on environmental conditions, but less so than species-species interactions; this is because they depend more on metabolic capabilities rather than on a variety of conditions in which species interact a particular way.

"Even though at first glance, GutCP appears similar to gap-filling during flux-balance analysis (FBA)" - please rephrase, I don't think anyone would think the approaches are similar.

We have re-written this paragraph to highlight the connection between both approaches.

Lines 474-484: *“GutCP is conceptually analogous to gap-filling during flux-balance analysis (FBA), but operates at a community level. Gap-filling infers intra-cellular metabolic reactions required for growth of a single microbial species in a particular medium. GutCP infers extra-cellular, cross-feeding interactions required to better predict the levels of several microbial species and metabolites simultaneously. Thus, one can think of GutCP as a community-level gap-filling: where each microbial species is effectively a net chemical reaction, and new cross-feeding interactions add new links between species”*

Consider placing the Methods before Results, I found it difficult to follow the Results at first. This might also be resolved by addressing minor point 3 above.

As suggested, we have addressed minor point 3.

"models that mapped to the 72 species and 221 metabolites in our dataset", and "we specifically extracted those reactions that were marked as extracellular, since those represented the consumption and production links that we are interested in" - please provide all such lists of reactions/metabolites in Supplementary Tables and refer to them.

We have provided the full list of these reactions as a new Supplementary Table 2, and referred to it appropriately.

"this model can predict the fecal metabolome with relatively high accuracy" - relative to what? Please specify/quantify.

We have removed the word “relatively”.

"We start with an initial cross-feeding network" - defined how? How many trophic layers? More details are needed.

We have added a few more sentences in the text providing more details about the initial cross-feeding network.

Lines 563-573: *“Each node in this bipartite network represented either a species or a metabolite. A directed link from a metabolite to a species denoted the ability of the species to consume the metabolite, and was termed a consumption link. Similarly, a link from a species to a metabolite indicated its ability to secrete the metabolite, and was termed a production link. While the network in the dataset was inter-connected and not hierarchical, we showed in a previous study that it could be roughly partitioned into four trophic levels, with the consumption of polysaccharides occurring at the top level, and the subsequent secretion of metabolic byproducts at later levels.”*

Reviewer #2 (Remarks to the Author):

This study proposes a computational approach that is claimed to be capable to identify novel and undetectable cross-feeding interactions within complex microbial ecosystem. Method has been explained in a well-organized and easy-to-understand manner, and reported results revealed potential of method to provide some mechanistic insight into microbial communities. However, there are some aspects that are needed to be explained more clearly or to be added to the study.

We thank the reviewer for their positive evaluation of our manuscript. We appreciated their feedback and suggestions, several of which led to revisions in the manuscript. We believe that these revisions have strengthened our results and increased the reliability of our method.

1.It has been mentioned that GutCP is generalizable and can be used with other mechanistic models, but authors preferred to use their own previously published consumer-resource model. To evaluate the power of the method, it would be helpful to employ at least one of the other mentioned models and present the results.

We appreciate the reviewer's concern about comparing results from different models. As suggested, we tried to run a popular method, that of flux-balance analysis (FBA) from Garza et al., but found that it is unfortunately rather computationally intensive. Running it for one individual was estimated to take roughly 40 days. Moreover, we are not experts in running such methods, and lack the computing resources. In light of this, we chose not to provide such a comparison in our manuscript.

2.Metabolic interaction between host and microbiome is an important factor in shaping the microbial communities and inter-species interactions. How and where in the method they include this factor.

We thank the reviewer for this comment to help clarify the assumptions of our method. There are a number of ways in which GutCP incorporates the host and its impact on the microbiome.

First, the length of the host's gut determines the number of trophic levels in our model of the gut microbiome. In a previous publication (Wang et. al., PLOS Comp. Biol. (2019)), we calibrated the number of trophic levels to the observed data in the gut, and found that 4 trophic levels were most consistent with the data. There, we also discussed that the number of levels represent the number of (nearly discrete) steps in which nutrient digestion occurs in the gut, which is in part controlled by the length of the host's gut. The variability in the number of trophic levels between data from different individuals reflects this difference (see Wang et. al.).

Second, the metabolome samples we used to test our model come from the middle of stool samples (not the surfaces). Metabolome measurements are typically taken from the middle of such samples because they are less sensitive to absorption by the host along the gut lining, and thus produce more consistent results with lower variability. By using such measurements, our model is somewhat buffered against the host's influence on the measurements.

We have now added new text in the Methods explaining how the effect of the host was incorporated for readers.

Lines 537-541: *“While we did not explicitly include the host in our model (for simplicity), we included the number of trophic levels which were consistent with gut -omics data. These trophic levels are in part determined by the length of the host’s gut.”*

3. When adding new cross-feeding interactions, how they define the directionality of the new links?

Our model explicitly defines interaction direction: either the production of a metabolite by a species, or its consumption. While adding new interactions, GutCP considered each direction separately, and calculated a prior probability on all interactions, based on whether the level of that metabolite was systematically under or over-predicted (described in Methods). It then proposed a new interaction and a new direction (production or consumption) randomly, with the aforementioned prior probability. For each species-metabolite pair, we thus have two possible links based on direction: those directed from species to metabolites represent production, while those directed from metabolites to species represent consumption. When both links are present, we term the pair a bidirectional link.

4. Authors evaluated the results using the systematic bias: the average deviation of the predictions from measured levels across all samples. Samples usually are collected in different conditions and species show different metabolic profiles in different conditions. How model solve the effect of outlier samples and changed metabolic profiles?

We understand the reviewer’s concern regarding how we treated outliers as well as accounting for the environmental dependence of metabolic interactions. We are happy to clarify both these concerns one by one.

First, GutCP does not aim to find interactions specific to human individuals, only to find interactions that are likely common and relevant to many individual gut microbiomes. We indicate this to readers in lines 151-167, where we highlight that GutCP focuses on the systematic bias across samples instead of individual sample biases. To avoid outliers greatly affecting systematic bias, we averaged the bias over all samples (N = 41) instead of, say adding them. One or two outliers in these measurements were thus unlikely to have a large effect on the systematic bias. Further, by taking the logarithm of deviations between measured and predicted values, we focused on changes in the bias at the order of magnitude level.

Second, as the reviewer correctly highlighted, species’ metabolic profiles may indeed depend on environmental conditions. While we do not explicitly account for such differences, GutCP prioritizes the discovery of species-metabolite interactions that systematically improve metabolome predictions across many samples. Thus, it is likely to detect those interactions that are less sensitive to environmental conditions, while ignoring others that are specific to certain conditions. The latter are likely to be ignored because it is unlikely for such interactions to significantly affect the systematic bias across many samples (Methods). Further, our results from cross-validation showed that GutCP was indeed able to find generalizable interactions using only a training dataset that also improved performance across an unseen test dataset (Table 1). This suggests that the interactions discovered by GutCP are likely those that are preserved across several contexts.

We thank the reviewer for this question, and now explain this aspect of GutCP for readers in the Discussion.

Lines 151-167: “For each metabolite and microbial species, there can be two kinds of prediction errors, or biases: individual (sample-specific difference between predicted and measured levels) and systematic (average difference across all samples). We focused on the “systematic bias” for each metabolite and microbial species: the average deviation of the predicted levels from the measured levels across all samples in our dataset (Fig. 1a, bottom). The systematic bias for each metabolite and microbe tells us whether our model generally tends to predicts their level to be greater than observed (over-predicted), less than observed (under-predicted), or neither (well-predicted). We assume that metabolites and microbes with a large systematic bias are most likely to harbor missing interactions \textcolor{red}{that are relevant across many samples.} We prioritize adding links to them in proportion to their systematic biases.”

Lines 484-501: “A key limitation of our method is that GutCP is built to reliably predict only those interactions which have a large and measurable impact on the gut microbiome and fecal metabolome profiles. Thus, it can miss interactions between rare species and metabolites, which might be real but not have a significant impact on metabolome predictions (GutCP works best when the improvement in predictions is of at least one order of magnitude). However, because GutCP prioritizes the discovery of species-metabolite interactions that systematically improve predictions across many samples, it is likely to detect those interactions that are less sensitive to environmental conditions, while ignoring others that are specific to certain conditions. Thus, the interactions predicted by GutCP are quite likely to be generally relevant in the gut microbiome.”

5.As the complex systems current and future states are sensitive to initial conditions, how model decides which new links to add first?

At every step, GutCP stochastically chooses which link to add next. The links are chosen from a prior probability distribution, which is calculated such that systematically over-predicted metabolites have a high probability of a new consumption link being added to them, while under-predicted ones have a high probability of a new production link being added to them (see Methods for complete details of this calculation). The rationale for this (as illustrated in Fig. 1, bottom) is that: metabolites that are consistently over-predicted by our model can yield correct predictions if more species consumed them.

And how it guarantees that starting from different links and following different trajectories will not result in biased predictions? How model tackle with risk of trapping in local optimum solutions and missing global optimum?

We cannot guarantee that we will reach the global optimum, but we use several machine learning and optimization methods to increase our chances of avoiding being trapped in any specific local optimum. For instance, we used an optimization procedure inspired by simulated annealing, where we started with a high search “temperature” and slowly reduced it over the procedure. When the temperature is high, the algorithm often escapes local minima by accepting links that can deteriorate performance, explores the space, and later, as the temperature reduces, it moves towards (hopefully the global) minimum by only accepting links that improve performance.

Another thing we did to avoid using interactions from any specific trajectory was that we simulated 100 optimization trajectories; we accepted only those interactions that were reproducibly detected at a high level of confidence ($P < 10^{-3}$; Fig. 2b).

6. Authors used test dataset from 41 human individuals, comprising 221 metabolites and 72 microbial species. How performance of the algorithm evaluated for larger communities consisting of higher number of species? especially knowing that gut microbiota consists of thousands of species and metabolites. It is required to perform sensitivity analysis based on size of the microbial community.

We agree with the reviewer's claim that the performance of the model may be a function of the number of species in the community. As suggested, we performed a sensitivity analysis, and encouragingly found that there was no correlation between model performance and the number of species in the community (Fig. S8). This suggests that it is unlikely for the model to be more or less reliable for communities with more species.

Even though microbiomes can consist of an even higher number of species, it is the most abundant species that will have the largest impact on the metabolome. These abundant species have always been included in our analysis (we cover, on average, a total abundance of 78% of the community).

We have added a new supplementary figure showing the results of this analysis (Fig. S8), and thank the reviewer for raising this point.

7. Authors found that the newly predicted interactions had both direct and indirect effects on metabolite levels. Implementing indirect effect is not explained well. How it has been calculated and what will be the consequence of biased indirect effect on predictions.

We agree with the reviewer that indirect effects were not explained well in the original manuscript. We have re-written this part of the manuscript to clarify what an indirect effect is, and now provide an example of the same. While we did not quantify indirect effects, we re-state here that indirect effects comprise any discovered links where GutCP improves the prediction for a metabolite without adding a link that produces or consumes it. The improvement in prediction comes entirely from other added links, which can increase or decrease the levels of microbes that produce (or consume) that metabolite.

Lines 356-373: *“Strikingly, we also observed several indirect effects of GutCP's predictions. Indirect effects comprise any discovered links where GutCP improves the prediction for a metabolite without adding a link that produces or consumes it. The improvement in prediction comes entirely from other added links, which can increase or decrease the levels of microbes that produce (or consume) that metabolite. For example, GutCP inferred no new consumption or production links for 5-Aminovalerate (Fig. 3), but adding other links (e.g., the consumption of putrescine by Clostridium difficile) increased the abundance of microbes producing 5-Aminovalerate. These microbes then produced more 5-Aminovalerate such that it was no longer under-predicted. Note that interactions such as these can only be inferred by causal and mechanistic models; this is because they alone can find such emergent, indirect effects of the microbiome on the metabolome.”*

8. Genome scale metabolic models (GEMs) have been used to validate the predictions, although authors earlier mentioned that GEMs are over estimating interaction. How they control over estimation? I recommend a validation on a new cohort or some in vitro experiments.

We understand and appreciate the reviewer's concern regarding GEMs over-estimating the number of interactions. We completely agree that *in vitro* experiments would be a good way to fully and accurately test our predictions, but as we discussed, these are outside the scope of our manuscript. In the absence of such experiments, as well as data from a new cohort (note: we are working with other groups who are

performing experiments yielding such data, to be reported in future studies), we relied on genome annotation-based evidence to support our results. Here, we controlled for over-estimation in the GEM dataset by comparing our results with a null model and calculating an associated P value, which we found was significant. Namely, the null model assumed that all interactions were equally likely to be discovered, and calculated the expected overlap between the predictions and GEM data. We normalized these calculations by the number of interactions estimated by the GEM data.

Finally, GSMM indeed over-estimate the number of interactions. To show this, we performed new simulations to compare the performance of our model when we use all GSMM-based exchange reactions with those of GutCP. We found that using all exchange reactions resulted in significantly lower performance (average Pearson correlation 0.26 compared with 0.74 for the optimal GutCP model).

Reviewers' Comments:

Reviewer #1:

Remarks to the Author:

I am happy with the edits and additional results which I agree have improved the manuscript. No further comments.

Reviewer #3:

Remarks to the Author:

I believe the authors' responses to reviewer #2's concerns and questions are satisfactory. Also, where needed they've run extra analyses and added extra explanations of clarifications to the manuscript. When reviewing new methods, my main concern usually is if it's tested on enough data and compared with enough number of similar methods in the field. However, in newer fields or fields with limited resources, it's not always possible to be thorough. The fact that the code is already active and available online, I believe will encourage more research and data production in the field.

On using GutCP with other mechanistic models (i.e. generalizability - although this is not the biggest advantage or selling point of this method), claiming it but not being able to show it in practice, for time-related or technical reasons, renders this claim pointless.

On the impact of the host on microbiome dynamics, I agree with the authors that adding the host to the model will greatly increase the complexity of the model and will make it almost intractable. Authors' efforts in including gut's trophic levels and minimizing the host impact by collecting from the middle of the stool are reasonable enough for this purpose.

On the directionality of cross-feeding links, the model's definition of interaction directions, production, and consumption for links originating from and ending in species, respectively, is plausible.

On systematic vs individual bias, authors have provided additional relevant explanations in the manuscript and clarified the limitations of the method.

On the initialization of trajectories (sensitivity to initial conditions) and the risk of being trapped in a local optimum, the authors provided a clear explanation and highlighted the satisfactory results of their simulation study.

On the impact of sample size on the algorithm's performance, the authors' explanation that the most abundant species have the largest impact on the metabolome and their dataset covered 78% of the total abundance of the community is sound. Furthermore, they performed a sensitivity analysis to prove their point that the increase in microbiome size would not hamper this algorithm.

On the prediction of the indirect effect of interactions on the metabolite levels, I believe the authors provide a clear and reasonable explanation and have modified the manuscript to reflect that.

On using a new cohort or in vitro experiments to control for GEMs overestimation, I agree that in vitro validation would be ideal but it's almost impractical for many groups. The lack of a similar cohort to validate the algorithm, however, is somewhat intriguing. Ideally, a novel method should be, along with similar methods, tested on a few simulated and real-life datasets to obtain the most robust measure of the performance of the novel algorithm. That said, I also understand that data production and tool development go hand in hand. Having more robust and practical tools will boost data production and research in the field.

REVIEWERS' COMMENTS

Reviewer #1 (Remarks to the Author):

I am happy with the edits and additional results which I agree have improved the manuscript. No further comments.

We thank the reviewer for their positive response.

Reviewer #3 (Remarks to the Author):

I believe the authors' responses to reviewer #2's concerns and questions are satisfactory. Also, where needed they've run extra analyses and added extra explanations of clarifications to the manuscript. When reviewing new methods, my main concern usually is if it's tested on enough data and compared with enough number of similar methods in the field. However, in newer fields or fields with limited resources, it's not always possible to be thorough. The fact that the code is already active and available online, I believe will encourage more research and data production in the field.

We thank the reviewer for their feedback, and agree with their assessment.

On using GutCP with other mechanistic models (i.e. generalizability - although this is not the biggest advantage or selling point of this method), claiming it but not being able to show it in practice, for time-related or technical reasons, renders this claim pointless.

We agree with the reviewer's comment. We have removed this claim for the manuscript.

On the impact of the host on microbiome dynamics, I agree with the authors that adding the host to the model will greatly increase the complexity of the model and will make it almost intractable. Authors' efforts in including gut's trophic levels and minimizing the host impact by collecting from the middle of the stool are reasonable enough for this purpose.

On the directionality of cross-feeding links, the model's definition of interaction directions, production, and consumption for links originating from and ending in species, respectively, is plausible.

On systematic vs individual bias, authors have provided additional relevant explanations in the manuscript and clarified the limitations of the method.

On the initialization of trajectories (sensitivity to initial conditions) and the risk of being trapped in a local optimum, the authors provided a clear explanation and highlighted the satisfactory results of their simulation study.

On the impact of sample size on the algorithm's performance, the authors' explanation that the most abundant species have the largest impact on the metabolome and their dataset covered 78% of the total

abundance of the community is sound. Furthermore, they performed a sensitivity analysis to prove their point that the increase in microbiome size would not hamper this algorithm.

On the prediction of the indirect effect of interactions on the metabolite levels, I believe the authors provide a clear and reasonable explanation and have modified the manuscript to reflect that.

We agree with all these comments, and thank the reviewer for them.

On using a new cohort or in vitro experiments to control for GEMs overestimation, I agree that in vitro validation would be ideal but it's almost impractical for many groups. The lack of a similar cohort to validate the algorithm, however, is somewhat intriguing. Ideally, a novel method should be, along with similar methods, tested on a few simulated and real-life datasets to obtain the most robust measure of the performance of the novel algorithm. That said, I also understand that data production and tool development go hand in hand. Having more robust and practical tools will boost data production and research in the field.

We understand and share the reviewer's concern — as we stated in a previous response, we are actively working with experimental collaborators to generate new data and test our method more rigorously.

Mehdi Layeghifard